# Bowhead whale faeces link increasing algal toxins in the Arctic to ocean warming

Kathi A. Lefebvre[1✉], Patrick Charapata[2], Raphaela Stimmelmayr[3], Peigen Lin[4], Robert S. Pickart[5], Katherine A. Hubbard[6], Brian D. Bill[1], Gay Sheffield[7], Emily K. Bowers[1], Donald M. Anderson[8], Evangeline Fachon[5,9] & Rick Thoman[10]

Over the last two decades, ocean warming and rapid loss of sea ice have dramatically changed the Pacific Arctic marine environment[1–3]. These changes are predicted to increase harmful algal bloom prevalence and toxicity, as rising temperatures and larger open water areas are more favourable for growth of some toxic algal species[4]. It is well known that algal toxins are transferred through food webs during blooms and can have negative impacts on wildlife and human health[5–7]. Yet, there are no long-term quantitative reports on algal toxin presence in Arctic food webs to evaluate increasing exposure risks. In the present study, algal toxins were quantified in bowel samples collected from 205 bowhead whales harvested for subsistence purposes over 19 years. These filter-feeding whales served as integrated food web samplers for algal toxin presence in the Beaufort Sea as it relates to changing environmental conditions over two decades. Algal toxin prevalences and concentrations were significantly correlated with ocean heat flux, open water area, wind velocity and atmospheric pressure. These results provide confirmative oceanic, atmospheric and biological evidence for increasing algal toxin concentrations in Arctic food webs due to warming ocean conditions. This approach elucidates breakthrough mechanistic connections between warming oceans and increasing algal toxin exposure risks to Arctic wildlife, which threatens food security for Native Alaskan communities that have been reliant on marine resources for subsistence for 5,000 years (ref. 8).

Warming sea surface temperatures (SSTs), the associated loss of sea ice quality, extent and duration as well as increases in open water area and duration in the Alaskan Arctic are predicted to cause increased growth of harmful algal bloom (HAB) species[4,9,10]. Two taxa of primary concern are *Pseudo-nitzschia* species (diatom) and *Alexandrium catenella* (dinoflagellate), which produce the potent neurotoxins, domoic acid (DA) and saxitoxin (STX), respectively[11]. These toxins accumulate in filter-feeding organisms such as clams, planktivorous fish and zooplankton (primarily euphausiids and copepods), which consume toxic algae and present substantial health risks to marine wildlife that consume contaminated prey[6,12,13]. Ingested toxins can cause potentially fatal human illnesses known as amnesic shellfish poisoning (ASP), caused by DA, and paralytic shellfish poisoning (PSP), caused by STX[14,15]. In the present study, these algal neurotoxins were quantified in faecal samples collected from the bowels of bowhead whales (*Balaena mysticetus*) landed during aboriginal subsistence harvests in the Beaufort Sea, Alaska, during the autumn (August to October) from 2004 to 2022 (*n* = 205 whales; Supplementary Table 1). Samples were collected as part of the bowhead whale harvest monitoring programme led by the North Slope Borough (NSB) leadership and the Alaska Eskimo Whaling Commission in collaboration with the 11 bowhead whaling communities in Alaska for health status assessments and tissue sample collections for baseline data on life history, natural diseases and marine threats[16,17]. Given their zooplankton diet and northern habitat, bowhead whales can also serve as integrated multidecadal in situ environmental biological samplers for the presence of algal toxins in Beaufort Sea food webs. These wide-ranging baleen whales filter-feed throughout the water column, primarily on copepods (*Calanus* sp.) and krill (euphausiids)[18], making them excellent sentinels for trophic transfer of algal toxins as it relates to climate variability over time[19]. In this study, data on prevalence and concentrations of algal toxins in bowhead whale faecal samples collected over the last two decades are used to demonstrate that specific shifts in Arctic environmental conditions related to climate change are causing higher risks of HAB toxin exposures to Arctic marine wildlife, ecosystems and to the peoples that rely on marine wildlife for nutritional, cultural and economic well-being.

## Prevalence and source of algal toxins

During 2004–2022, faecal sample collections were analysed for yearly prevalence of algal toxins from 205 bowhead whales that had been

[1]Environmental and Fisheries Sciences, Northwest Fisheries Science Center, NOAA Fisheries, Seattle, WA, USA. [2]Center for Species Survival, Georgia Aquarium, Atlanta, GA, USA. [3]Department of Wildlife Management, North Slope Borough, Utqiaġvik, AK, USA. [4]School of Oceanography, Shanghai Jiao Tong University, Shanghai, China. [5]Physical Oceanography Department, Woods Hole Oceanographic Institution, Woods Hole, MA, USA. [6]Fish and Wildlife Research Institute, Florida Fish and Wildlife Conservation Commission, St. Petersburg, FL, USA. [7]Alaska Sea Grant/Marine Advisory Program, University of Alaska Fairbanks, Nome, AK, USA. [8]Biology Department, Woods Hole Oceanographic Institution, Woods Hole, MA, USA. [9]Department of Earth Atmospheric and Planetary Sciences, Massachusetts Institute of Technology, Cambridge, MA, USA. [10]International Arctic Research Center, University of Alaska Fairbanks, Fairbanks, AK, USA. ✉e-mail: Kathi.Lefebvre@noaa.gov

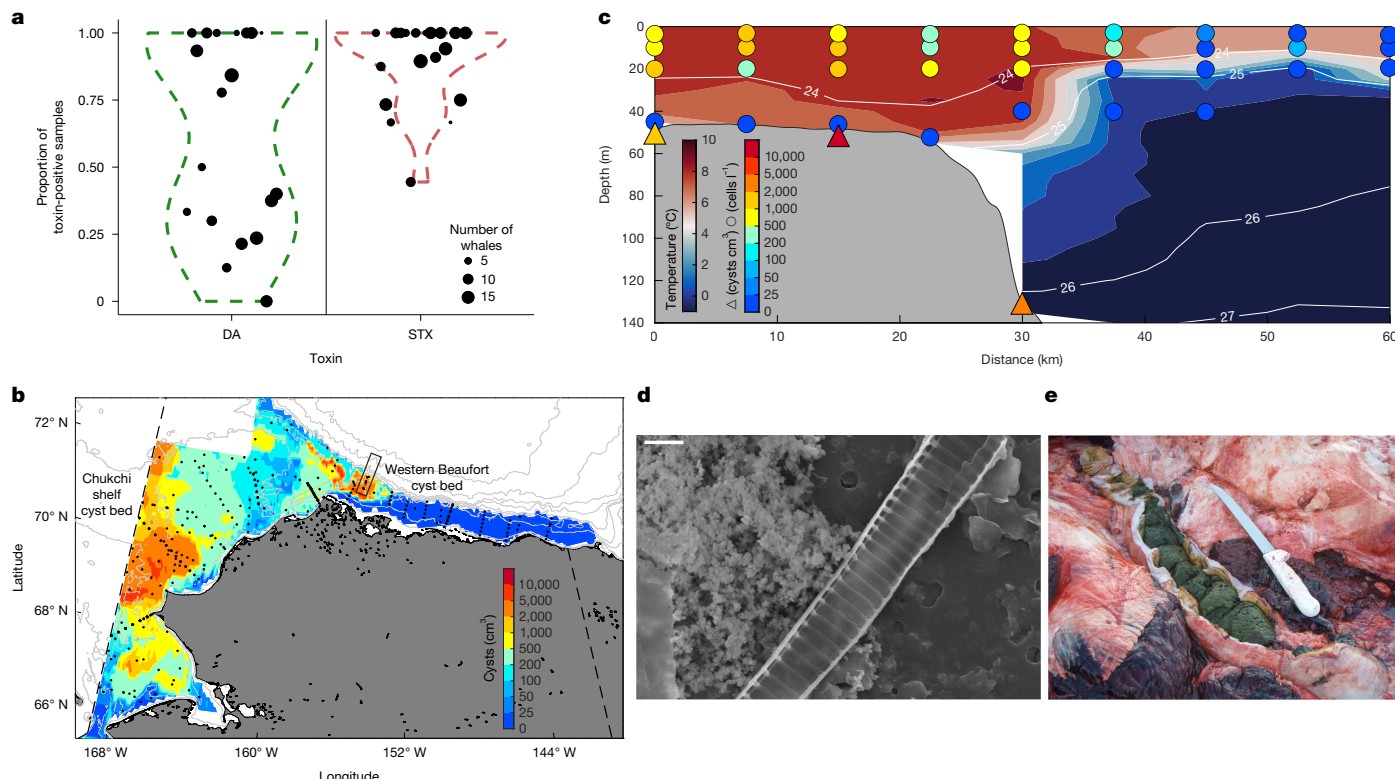

**Fig. 1 | Algal toxins and species. a**, Prevalence of DA and STX in faecal samples from bowhead whales ($n = 205$) harvested for subsistence purposes during autumn of 2004 to 2022. Each point represents the proportion of whales that tested positive of the total whales sampled for that year depicted by point size ($n = 3–19$ whales per year). Horizontal distribution in points is an added jitter effect for effective visualization of similar toxin prevalences among years (points). Dashed lines are violin plots that visualize the distribution of prevalence data for each algal toxin (green for DA and red for STX). **b**, Distribution of *A. catenella* cyst beds on the Chukchi shelf and in the western Beaufort Sea, aggregated from samples collected from 2018 to 2022. **c**, Cross-sectional view of an *A. catenella* bloom detected in August 2019 over the western Beaufort cyst bed (see rectangle in **b**, looking along shore towards the west). Circles indicate cell concentrations throughout the water column (cells l⁻¹; Supplementary Table 2) and triangles indicate underlying cyst densities in the sediment (cysts cm⁻³). Background colour displays water temperatures, which were anomalously warm during this event, and density contours are overlaid in white (kg m⁻³). **d**, SEM image showing a partial *P.* cf. *seriata* frustule isolated from faeces. Scale bar, 2 μm. **e**, Photo of a dissected bowhead bowel during faecal sample collection.

feeding in the Beaufort Sea. Prevalence results ranged from 0% to 100% for DA and 44% to 100% for STX, showing that STX has a higher prevalence than DA in Beaufort Sea food webs and was present in nearly half or more of the whales sampled in all years (Fig. 1a). High STX prevalence is consistent with recent studies documenting increased risks of *A. catenella* blooms in Arctic waters due to warming conditions[4,20,21]. Comprehensive benthic sediment sampling throughout Arctic or subarctic oceans in recent years has found large accumulations of *A. catenella* resting cysts (cyst beds) broadly present in Arctic sediments, with notable accumulations on the Chukchi shelf as well as in the western Beaufort Sea[22,23] (Fig. 1b). Germination activity of these *A. catenella* cyst beds is tightly coupled with temperature[4,24], leading to elevated bloom risks in years with warm bottom waters. For example, a bloom of *A. catenella* was observed during summer 2019 concurrent with anomalously warm shelf bottom waters (7–9 °C) (Fig. 1c). At these temperatures, germination would occur in 10 days or less, whereas temperatures of 1–2 °C observed at the same location in other years would take 42–85 days (ref. 4). Additionally, vegetative *A. catenella* cells in warmer surface waters have increased growth rates[25,26]. Therefore, *Alexandrium* blooms observed on the Beaufort shelf[10] can arise through two potential mechanisms: (1) advection of *A. catenella* populations that originate in the Bering or Chukchi Seas and/or (2) local germination from the dense *A. catenella* cyst bed found in the western Beaufort shelf sediments. Both mechanisms are influenced by warmer temperatures[4] (Fig. 1b).

Several species of *Pseudo-nitzschia* known to produce DA are present in the Arctic, where species often co-occur in mixed assemblages[27].

*Pseudo-nitzschia* species composition in the Beaufort Sea can include toxic subpolar and/or polar species of *Pseudo-nitzschia*. Potentially toxic *Pseudo-nitzschia* species or populations in the Beaufort Sea can include those considered endemic to the Arctic, including some observed in sea ice[27]. Similarly, toxic species with distributions spanning southern temperate to subpolar Pacific waters have been documented in the Beaufort Sea. In the present study, one sample of bowhead faeces collected in autumn 2017 was found to contain 6,259 ng DA g⁻¹ and frustules isolated from this sample were identified as *Pseudo-nitzschia* cf. *seriata* by means of scanning electron microscopy (SEM; Fig. 1d,e). *P.* cf. *seriata* is a particularly toxic species[28], which has at least two genetically distinct populations that occupy the Arctic[27]. One population seems to be polar and has been observed in upwelling zones along the eastern Beaufort shelf[27]; a separate *P. seriata* genotype was previously recorded in temperate and subpolar Pacific waters as well as the Bering Strait and Chukchi Sea[29], making it difficult to assess the origin of the frustules in question. Other taxa observed in whale faeces by means of SEM included *Pseudo-nitzschia pungens*, a less toxic but broadly distributed species, and *Pseudo-nitzschia obtusa*, a polar species previously considered to be non-toxic until toxicity was induced through exposure to copepods[27] (Extended Data Fig. 1). These limited results allow the development of separate, but not mutually exclusive, hypotheses about trophic transfer of DA: (1) toxic Pacific *Pseudo-nitzschia* assemblages advected from the Northern Bering and Chukchi Seas are present (and thus consumed) in the Beaufort Sea; (2) endemic toxic polar *Pseudo-nitzschia* assemblages are present (and thus consumed) in the Beaufort Sea; and (3) toxic Pacific and polar assemblages may co-occur in the Beaufort Sea.

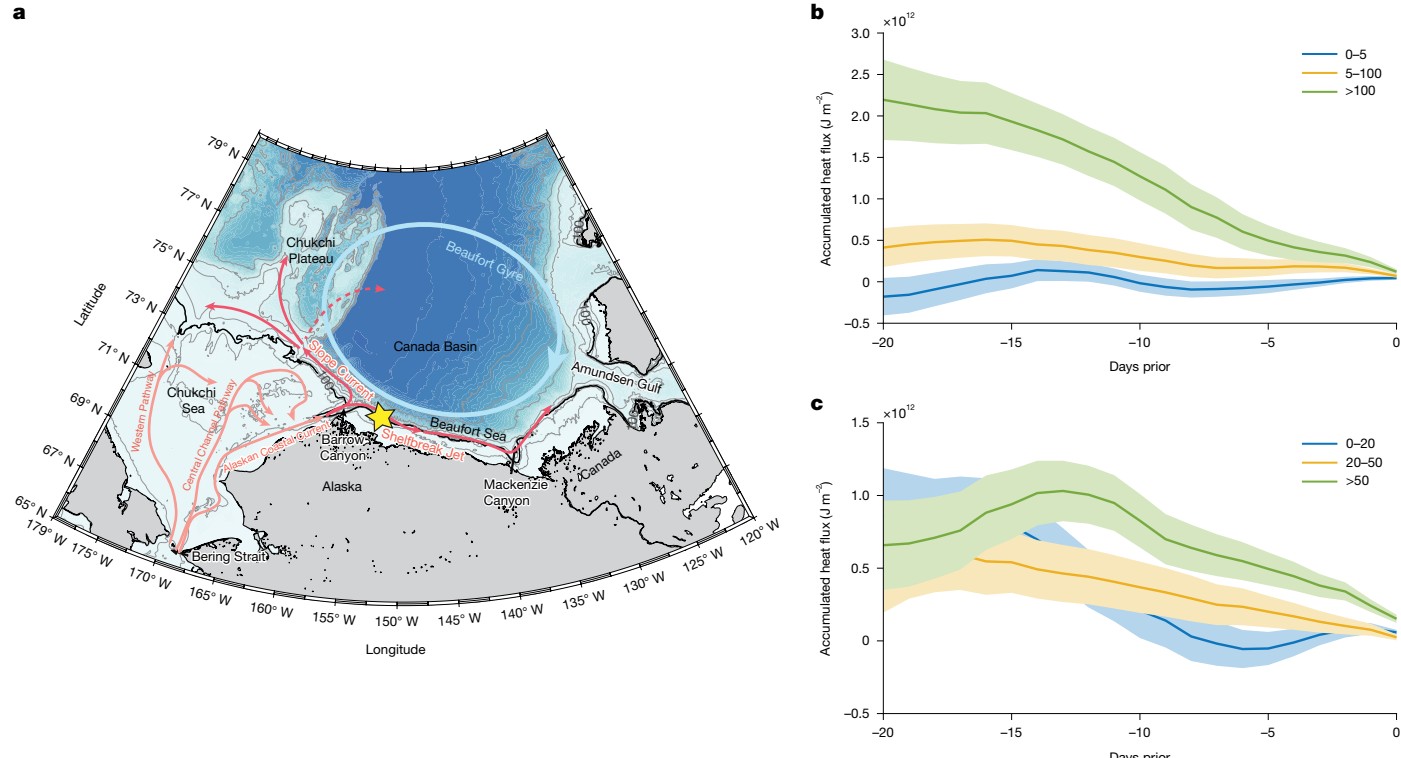

**Fig. 2 | Algal toxins and accumulated heat flux. a**, Map of currents and location of the long-term oceanographic mooring (gold star). **b**, Composites of accumulated heat flux (J m$^{-2}$) measured at the mooring over the 20 days before bowhead whale faecal sampling for the three DA concentration groups (ng g$^{-1}$):

low ($n = 90$, blue), medium ($n = 73$, yellow) and high ($n = 42$, green). **c**, Same as **b** except with STX concentration groups (ng g$^{-1}$): low ($n = 80$, blue), medium ($n = 63$, yellow) and high ($n = 62$, green). The shading represents the standard error (s.d./√$N$) of the average heat flux for each composite.

## Toxin concentrations linked to heat flux

A mooring continually maintained since 2002 roughly 150 km east of Point Barrow in the Beaufort shelfbreak jet, provides water temperature and horizontal velocity data at a water depth of about 50 m. The Beaufort shelfbreak jet is the main conduit for Pacific water travelling from the Chukchi Sea eastwards to the Beaufort Sea (Fig. 2a). These data were used to compute the horizontal heat flux per unit area, hereafter referred to simply as heat flux (relative to the freezing point of the saltiest water flowing through Bering Strait, −1.91 °C; ref. 30), where positive is directed southeastward along the main path of the current. To quantify the effect of the heat flux on HAB toxin levels in bowhead whales, we considered three groups of DA concentrations from low to high values: 0–5, more than 5–100 and more than 100 ng DA g$^{-1}$ faeces (the results are not sensitive to the exact bin sizes). For each DA sample ($n = 205$) we accumulated the heat flux backwards in time by 20 days (the approximate advective time from Barrow Canyon to the feeding area) and then computed the composite accumulated heat flux for the three DA concentration groups (Fig. 2b). Over the full 20 days, the low DA group (blue curve) is associated with the lowest accumulated heat flux and the middle DA group is associated with slightly higher accumulated heat flux. The high DA group is associated with the highest heat flux, with increasing difference from the other two groups beyond 5 days prior. This indicates that higher DA concentration in whales is associated with enhanced heat flux from the Chukchi Sea to the Beaufort Sea, presumably because the warmer water is more conducive for bloom initiation and development. The analogous calculations were performed for three STX concentration groups: 0–20, more than 20–50 and more than 50 ng STX g$^{-1}$ of faeces (Fig. 2c). The results show that within 10 days prior, the high STX group (more than 50 ng g$^{-1}$) is associated with the highest heat flux among the three groups, whereas the two smaller groups were mostly indistinguishable over this time period.

These results indicate a clear relationship between upper layer heat flux and HAB toxin concentrations in bowhead whales and confirm that warmer ocean conditions are linked to higher HAB toxin loads in the food web. Both advected vegetative *A. catenella* cells from southern waters and cells germinated from local benthic cyst beds, are potential sources for STX-producing blooms in the Beaufort Sea, whereas *Pseudo-nitzschia* lacks the benthic contribution as it does not produce cysts. This may explain why DA has a stronger relationship to the heat flux in the 20 days before bowhead harvest, which reflects the advective time from Barrow Canyon to the feeding area near the mooring site. By contrast, STX concentrations correlate with heat flux within 10 days of bowhead harvest. This shorter time frame is probably due to the contribution of the western Beaufort Sea cyst bed located closer to the feeding site east of Point Barrow, in addition to advected *A. catenella* cells and probably explains the higher prevalence and potential toxicity of STX in Arctic food webs compared to DA (Fig. 1b). Further comparisons with standard Beaufort Sea summer SST anomalies and toxin concentrations found significant correlations for DA, but not for STX also probably due to the already higher prevalence of STX in the Beaufort Sea (Extended Data Fig. 4).

## Wind and pressure linked to algal toxins

To investigate how ocean heat flux might be linked to the atmospheric forcing for each DA and STX concentration grouping, we constructed wind composites in the study region. To be consistent with the timing of the heat flux composites, the wind composites represent an average over the 15- and 10-day period before when each faecal sample was collected for the three DA groups and STX groups, respectively (results are not sensitive to the precise averaging period). For both the low DA and low STX concentration groups, the Beaufort Sea region was characterized by strong northeasterly winds, whereas the moderate

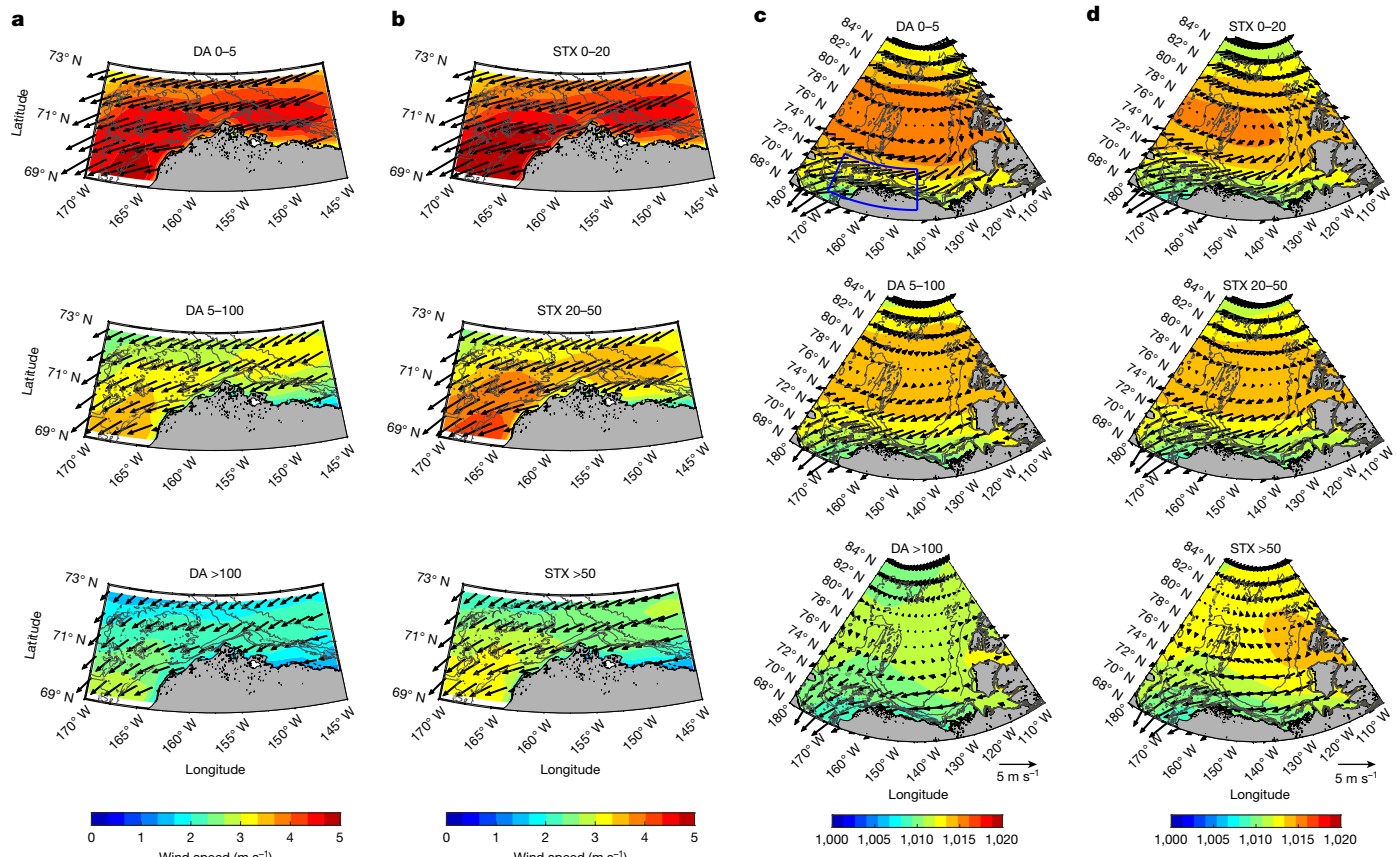

**Fig. 3 | Wind velocity and SLP. a–d**, Composites of wind vectors and wind speed (colour scale) for the three concentration groups for DA (**a**) and STX (**b**) and corresponding composites of SLP (colour scale) and wind vectors over a larger domain for the three concentration groups for DA (**c**) and STX (**d**).

DA concentration groups (ng g⁻¹) are low (0–5), medium (>5–100) and high (>100); STX concentration groups (ng g⁻¹) are low (0–20), medium (>20–50) and high (>50). The blue box in **c** denotes the enlarged domain in **a** and **b**.

and high concentration groups were characterized by progressively weaker winds, corresponding to times of faecal collections (Fig. 3a,b). It is well known that strong northeasterly winds weaken or even reverse the Pacific water outflow from Barrow Canyon[31–33], which is consistent with less heat flux towards the Beaufort Sea. The local winds are typically tied to large-scale atmospheric systems, in particular the Beaufort High at this time of year[34]. Analogous composites of sea-level pressure (SLP) over a broader domain show that the Beaufort High is strongest for the low DA and low STX groups and, when weakened, the toxin concentrations increased (Fig. 3c,d). Our results indicate that an enhanced Beaufort High leads to stronger northeasterly winds in the study region, which weaken the outflow from Barrow Canyon and decrease the heat flux to the Beaufort Sea. The combination of reduced advection of blooms from the south and slower growth rates due to decreased heat flux results in lower bloom densities and therefore lower toxin levels in zooplankton ingested by bowhead whales as they traverse the region.

## Open water area linked to DA prevalence

Further evidence for the role of climate change-related Arctic warming to increased DA presence is shown in the relationship between Beaufort Sea open water area anomalies and DA prevalence in whales. Anomalies of open water (km²) in the Alaskan Beaufort Sea were calculated by dividing summer monthly open water values for each year of the study period (2004–2022) by their respective averaged 1982–2011 summer month open water (km²) baseline values. Larger open water anomalies were observed in months leading up to and during the autumn

bowhead harvest seasons in which DA prevalence in whales was 100% (Fig. 4a). In all years that DA was present in 100% of bowhead whales sampled, open water area anomalies were significantly higher in June, July, August and September, than in years with less than 100% DA prevalence (Fig. 4a and Supplementary Table 3). Similar trends were observed in comparisons with less than 90% and less than 75% prevalence cutoffs (Extended Data Fig. 2a and 2b, respectively). The greatest difference in open water area between prevalence groups was observed during June (Fig. 4a and Extended Data Fig. 2). Additionally, June open water area is positively correlated with SST departure from baseline in the following month of July (Fig. 4b). These findings indicate that increased and earlier open water area in June, along with associated warming water during July, sets the stage for earlier and more favourable conditions for *Pseudo-nitzschia* blooms, resulting in higher DA prevalence in Beaufort Sea food webs during corresponding autumn bowhead whale harvest seasons. This correlation was also seen for STX prevalence, but at a lower magnitude (Extended Data Fig. 3).

## Implications for Arctic food webs

Our data show direct impacts of ocean warming and reductions of annual sea ice extent on the prevalence and concentration of HAB toxins in Beaufort Sea food webs over nearly two decades by means of integrative ecosystem biosampling by filter-feeding bowhead whales. Two separate sources for *A. catenella* blooms, (1) cells advected in surface waters from the Bering and Chukchi Seas and (2) local germination of cysts from the cyst bed east of Point Barrow (Fig. 1b), explain the higher prevalence of STX observed in Beaufort Sea food webs compared to

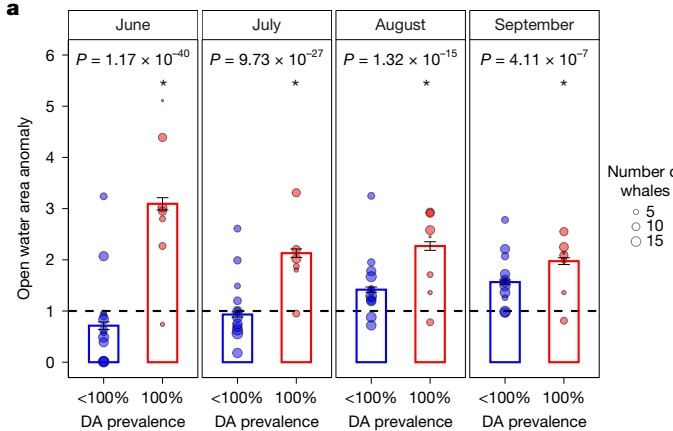

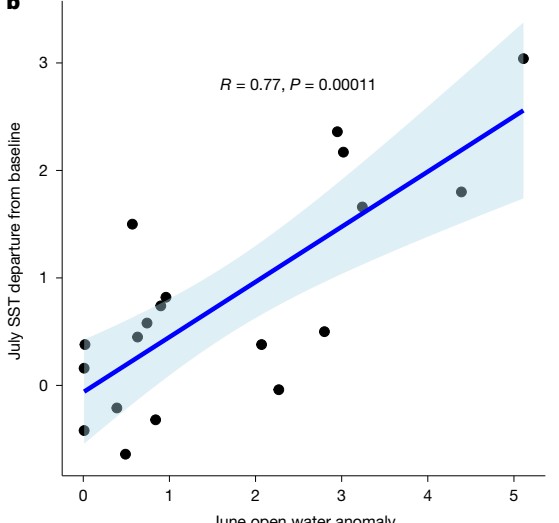

**Fig. 4 | Open water anomalies. a**, Comparison of estimated marginal means (±s.e.) of Beaufort Sea open water area anomalies in June, July, August and September between years with 100% DA prevalence (red bars and points; $n = 56$ whales sampled across $Y = 7$ independently sampled years) and less than 100% DA prevalence (blue bars; $n = 149$ whales sampled over $Y = 12$ independently sampled years) in bowhead whales. Size of points represent the total whales sampled for that year ($n = 3$–19 whales per year). Open water anomalies are calculated by dividing monthly open water areas ($km^2$) by the monthly baseline areas ($km^2$), resulting in a unitless anomaly equating baseline open water area values to 1 (dashed black line). *Open water anomalies were significantly higher in years with 100% prevalence (unpaired two-sided $t$-test with no $P$-value adjustments; Supplementary Table 3). **b**, Positive correlation ($R = 0.77$, $P = 0.00011$, $n = 19$ years) between June open water area anomalies and July SST departure from baseline anomalies (baseline 1982–2011) for 2004–2022 with the blue line representing the best linear fit of the data and shaded region representing the corresponding 95% confidence interval.

DA (Fig. 1a), whereas *Pseudo-nitzschia* species are more dependent on advective processes for introduction into the Beaufort Sea. Warmer ocean temperatures increase the rates of both *A. catenella* cell growth and cyst germination resulting in larger more toxic blooms[4] (Fig. 1c). Dangerously high STX concentrations have recently been documented in Arctic food webs[13]. By contrast, DA prevalence is lower than STX and DA concentrations quantified in bowhead whale faeces are considered low in terms of poisoning risks to bowhead whales. The concern, however, is the future expansion of DA and DA-producing blooms with continued Arctic warming and sea ice loss. This trend will probably continue as Arctic/Subarctic seas continue to warm at an accelerated rate (Fig. 5). Long-term trend data for SSTs (NOAA-ERSST v.5)[35] and sea ice (NSIDC-SII v.3)[36] show multidecade SST warming since 1900 and

concurrent decreases in summer sea ice extent in the Bering, Chukchi and Beaufort Seas (Fig. 5). Additionally, these data show acceleration of both ocean warming and sea ice loss trends in the last two decades (Fig. 5). Since 1900, the 10-warmest summer average SSTs in the Bering and Chukchi Seas have been after 2000 (Fig. 5a,b). A similar pattern is shown by the accelerating reduction of sea ice extent from 1979 to 2023 (Fig. 5d). Our findings from bowhead whale faecal samples provide statistically significant and mechanistic oceanic, atmospheric and biological evidence for increasing HAB toxin exposure risks in Beaufort Sea food webs due to conditions associated with climate change. Marine resources have been essential for the nutritional, cultural and economic well-being of northern and western Alaskan coastal communities for more than 5,000 years (ref. 8). Continued harvest monitoring for HAB toxin exposure in marine mammal sentinels such as bowhead whales is essential to ensure food safety and food security for northern peoples reliant on the marine food web.

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

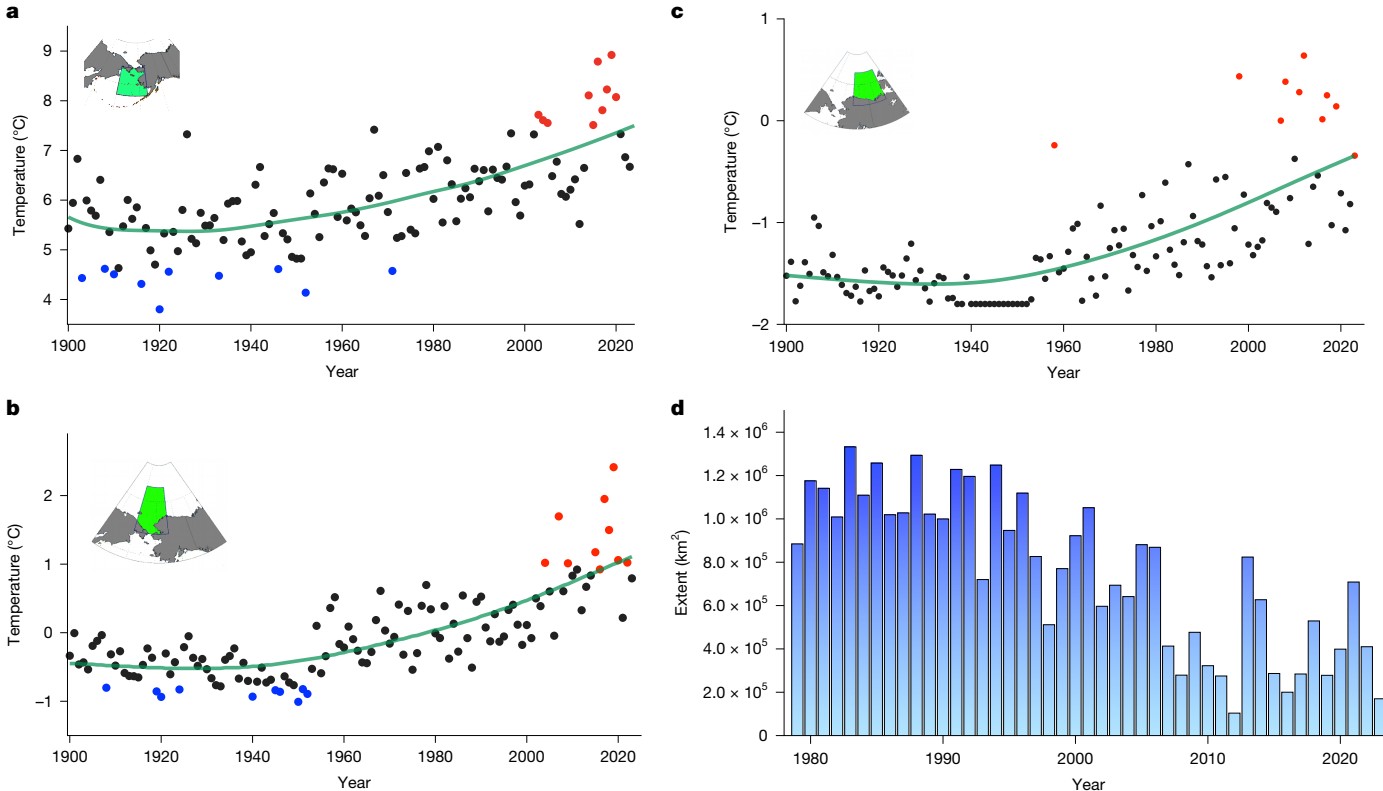

**Fig. 5 | Warming SST and minimum sea ice extent. a–c**, Average SST during May to September from 1900 to 2023 in the North Bering Sea (**a**), Chukchi Sea (**b**) and Beaufort Sea (**c**). Green line shows running average. Red dots are the 10 warmest years. Blue dots are the 10 coldest years. **d**, Bar graph of annual minimum sea ice extent in the Bering, Chukchi and Beaufort Seas from 1979 to 2023.

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

# Methods

## Sample and data collection

To define the relationship between warming climate conditions and algal toxin prevalence and concentration in Arctic food webs, we used the following data: algal toxin concentrations in bowhead whale faeces, *A. catenella* cell and cyst densities, presence of *Pseudo-nitzschia* frustules in bowhead whale faeces, ocean heat flux, wind velocity, SLP, open water area, SSTs and annual minimum sea ice extent.

## Bowhead whale faecal sample collection

During 2004–2022, faecal samples were collected from 205 bowhead whales harvested for subsistence purposes and landed at Utqiaġvik (formerly Barrow), Alaska, during autumn harvest seasons (August–October; Supplementary Table 1). The whales are known to feed near Utqiaġvik at depths ranging from shallow continental shelf (45 m) to the deeper waters (more than 300 m) of Barrow Canyon[18]. Depths at which *Alexandrium* cells mainly occur are in the upper 25 m (ref. 4) and *Pseudo-nitzschia* particulate DA measurements have been documented from the surface (about 2 m) to chl-*a* maximum depths (20–40 m)[27]. Whales are typically harvested within a 30-mile radius of Utqiaġvik[37]. Sections of colon from each whale were dissected and faecal matter was removed using plastic spoons. Samples were stored frozen in Whirl-Pak bags at −20 °C until analysed for algal toxins.

## Quantification of algal toxins in faecal samples

Algal toxins were extracted from frozen bowhead whale faecal samples (*n* = 205; Supplementary Table 1) by slowly thawing and stirring cold faecal material followed by subsampling into approximately 1 g for analysis. To each aliquot, 50% methanol was added at a volume of three times the aliquot weight for a one-in-four dilution. Samples were vortexed briefly on high (Analogue Vortex Mixer, sn 060223013, VWR) and homogenized using a generator probe (GLH 850, Omni-International) for 1 min at 2,100 rpm. Homogenates were then centrifuged at 4,100 rpm for 20 min at 4 °C (CR3i centrifuge, Jouan) and supernatants were poured off and stored in 4-ml amber glass vials in a dark refrigerator (about 1 °C). Directly before analysis, 200 μl of each sample were filtered through 0.22-μm Ultra-Free Centrifugal filters (UFC30GVNB, Millipore Sigma) in a tabletop centrifuge (AccuSpin Micro 17, Fisher Scientific) at 12,000 rpm. Methanol (50%) is the standard extraction solvent for DA enzyme-linked immunosorbent assay (ELISA) analyses[38] and has also been validated as an effective extraction solvent for STX ELISA analyses[39]; hence, these 50% methanol extracts were used for both DA and STX quantifications.

Algal toxins were quantified in bowhead whale faecal samples by means of commercially available direct-competition ELISA kits. DA was quantified using Biosense ASP ELISA kits (PN A31300401, Biosense Laboratories) for samples collected in 2004–2021 and using the comparable ABRAXIS Domoic Acid ASP ELISA kits (PN 520505, Gold Standard Diagnostics) for samples collected in 2022. STX was quantified using ABRAXIS Saxitoxins PSP ELISA (PN 52255B, Gold Standard Diagnostics) for all samples. Although these kits are designed to analyse shellfish and water samples, previous studies have determined appropriate dilutions to avoid matrix effects from marine mammal matrices and have validated ELISA results compared to other analytical methods[12,40,41]. Kits were used according to manufacturer's instructions with dilution modifications from ref. 41 and ref. 12 for DA and STX, respectively. Sample extracts were diluted 1:100 for DA and 1:50 for STX (sample to kit-provided sample diluent). Standards, controls, blanks, samples and kit-provided reagents were then added to ELISA plates in duplicate and processed following the kits instructions. Toxin quantifications were obtained using a BioTek Epoch plate reader (sn 257814) and concentrations (ng of toxin per g of sample) were determined using a four-parameter logistic curve model based on the known standards concentrations. Samples with concentrations exceeding the detection range of the kit (defined as 20–80% of the standards range) were diluted further and reanalysed until concentrations fell within the detection range. Minimum assay detection limits were 4 ng g$^{-1}$ for DA and 4.7 ng g$^{-1}$ for STX. All faecal samples collected from 2010 to 2022 were analysed within the year they were collected. Faecal samples from 2004 to 2009 were analysed within 5 years of collection. To rule out any potential toxin degradation issues, studies were performed with bowhead faecal material stored over 4 years under various storage conditions. Results from both STX and DA storage studies confirmed that long-term frozen storage did not impact toxin concentrations over time[39,42]. In several years of bloom sampling across the region during this project (2019, 2022 and 2023), the suite of toxins produced by Pacific Arctic *A. catenella* has been consistently dominated by gonyautoxin-4, neosaxitoxin, gonyautoxin-3 and STX[13]. Unfortunately, both gonyautoxin-4 and neosaxitoxin have low cross-reactivities with the ELISA test (less than 2%), but STX is picked up at 100% and gonyautoxin-3 at 23%. So, although the ELISA is probably underestimating the total amount of toxin in these faecal samples, the overall consistency observed in toxin profiles of regional *A. catenella* strains across years indicates that ELISA data are appropriate for assessing relative temporal trends in toxicity, such as the results reported in this study. These STX quantifications are representative of prevalence and relative concentrations equally over the two decades of sampling.

## *A. catenella* cyst sample collections

To map *A. catenella* cyst abundance in the study region, surface sediment samples were collected during 12 different cruises over 5 years (2018–2022; Supplementary Table 4). Sediments were collected using a Van Veen or Smith-McIntyre grab, and a cut syringe was used to collect a plug from the 0–3-cm layer of each grab; in some cases, several plugs were collected and pooled together. Each subsample was homogenized, sealed in an airtight container and maintained in the dark at 0–4 °C.

## Cyst microscopy and mapping

All sediment samples were processed using a primulin stain (Extended Data Fig. 1), allowing *A. catenella* cysts to be enumerated following established methods[43]. Briefly, an aliquot of each homogenized sediment sample was diluted (1:5) in filtered seawater and sonicated (Brandon Sonifier 250, 40% amplitude, 60 s) in an ice bath. The resulting slurry was sieved to isolate the 20–80-μm size fraction, which was then resuspended in filtered seawater and preserved with 5% formalin. The formalin-preserved samples were chilled (4 °C) for 1–3 h, after which they were centrifuged (3,000*g*, 10 min), formalin–seawater supernatant was aspirated and sediment pellet was resuspended in chilled methanol. After refrigeration (4 °C) in methanol for at least 72 h, the samples went through a series of centrifugation and aspiration steps to transfer from methanol to deionized water and from deionized water to primulin stain (2 mg ml$^{-1}$, 2 ml per sample). Samples were rotated on a Labquake for 1 h at 4 °C, after which more centrifugation steps were used to wash the sample in deionized water and resuspend each sample in a final volume of 10–15 ml of deionized water.

*A. catenella* cysts were enumerated using a Zeiss Axioscope epifluorescence microscope equipped with a FITC filter set (Zeiss 09, excitation 450–490 nm band pass; emission 515 nm long pass) under a ×10 objective. Samples that were too dense to count were diluted 1:10; all raw counts were normalized to *A. catenella* cysts per cubic centimetre. Results from all cruises were compiled together and a map of *A. catenella* cyst abundance (Fig. 1d) was produced in Matlab (R2024a) using the m_map package and an interpolation method which weights along isobaths[44] and which has been previously used to map cyst distributions in the region[4].

## *A. catenella* vegetative cell sample collections

During the HLY1901 cruise (Supplementary Table 4), vegetative cell concentrations of *A. catenella* were quantified in discrete water

samples collected and preserved at process stations. These samples were used to detect and characterize a bloom of *A. catenella* on the Barrow Canyon East transect line on 20 August 2019 (Supplementary Table 2). At each station, conductivity–temperature–depth (CTD) data were collected using a Sea-Bird 911 plus CTD mounted to a 24-position rosette[45]. Discrete 2-l water samples were collected from Niskin bottles representing surface (about 3 m) and 10-m, chlorophyll maximum and bottom depths. These samples were immediately sieved through a 15-μm Nitex mesh; all captured particles were backwashed with filtered seawater into a 15-ml conical tube and fixed with formalin (5% final concentration). Water samples were stored at 1 °C for up to 72 h, at which point they were centrifuged (3,000*g*, 10 min) and overlying seawater–formalin mixture was aspirated. The phytoplankton pellet was resuspended in chilled methanol and all samples were stored at −20 °C.

### Vegetative cell microscopy and enumeration
*A. catenella* were enumerated in preserved seawater samples using a fluorescence in situ hybridization method following previously published procedures[46]. Briefly, an aliquot of each methanol-preserved sample was transferred to a filtration manifold column fit with a 5.0-μm pore size, 25-mm diameter Cyclopore membrane filter. Vacuum suction was used to remove the methanol from each manifold column and replace it with 1 ml of prehybridization buffer (5× SET (750 mM NaCl, 5 mM EDTA, 100 mM Tris-HCl, pH 7.8), 0.1 μg ml⁻¹ of polyadenylic acid, 0.1% IGEPAL CA-630, 10% formamide). After a 5-min room-temperature incubation period, the prehybridization buffer was removed and replaced with 1 ml of hybridization buffer (prehybridization buffer augmented with 4.8 μg ml⁻¹ of Cy3 NA-1 probe). The NA-1 oligonucleotide probe (5′ Cy3-AGT GCA ACA CTC CCA CCA-3′) was selected to label *A. catenella* large subunit ribosomal RNA. The samples were then incubated in the dark (50 °C, 1 h), after which the hybridization buffer was removed and replaced with 1 ml of wash buffer (0.2× SET) for an extra 5-min room-temperature incubation. All remaining buffer was removed by means of vacuum filtration and filters were mounted on slides with a small volume (20–40 μl) 80% glycerol 25× SET solution. Samples were stored in the dark at 4 °C for up to 3 days before enumeration. In each sample, all *A. catenella* vegetative cells were enumerated at ×10 on a Zeiss Axioscope M1 using a Cy3 filter set (Chroma no. 49016/TRITC long pass); these cell counts were then normalized to cells per litre to determine in situ concentrations.

### *Pseudo-nitzschia* frustules in faecal samples
Subsamples of bowhead whale faecal samples (about 0.05–0.2 g) were prepared for SEM using published methods[47]. Briefly, faecal samples were rinsed three times with 1 ml of distilled water in 1.5-ml microcentrifuge tubes. Each rinse step included vortexing and centrifugation of the pellet. Pellets were then oxidized for 2 h with four or five drops of saturated potassium permanganate solution, cleared with three rinses of concentrated HCl, rinsed again three times with distilled water and then filtered onto 13-mm diameter, 1.2-μm pore size, polycarbonate filters (Millipore Corp). Filters were then glued to aluminium stubs, coated with gold-palladium and viewed in a JEOL 6360LV SEM. Species determinations were made using published morphological characteristics[48] (Extended Data Fig. 1).

### Mooring data and heat flux calculation
We used the hydrographic and velocity data from a mooring close to Barrow Canyon, maintained since 2002, as part of Arctic Observation Network[49,50]. To compute the lateral heat flux in the upper layer of the Beaufort shelfbreak jet, we used the temperature and velocity data measured at the top float of the mooring (approximately 35-m depth). The accumulated lateral heat flux (*H*) is computed as:

$$H = \sum_t \rho C_p (\theta(t) - \theta_r) u(t), \tag{1}$$

where $\rho$ is potential density, $C_p$ is the specific heat of seawater, $\theta(t)$ is the time-dependent potential temperature, $\theta_r = -1.91$ °C is the reference temperature and $u(t)$ is the time-dependent alongstream component (125° clockwise from north) of the velocity. The heat flux was averaged for each composite toxin concentration group of bowhead whales (low DA $n = 90$, STX $n = 80$; moderate DA $n = 73$, STX $n = 63$; high DA $n = 42$, STX $n = 62$).

### Wind velocity and SLP reanalysis
We used the hourly wind field and SLP from the ERA5 reanalysis, with a spatial resolution of 0.25°, provided by the European Centre for Medium-Range Weather Forecasts[51]. The ERA5 data have shown good agreement with observations in the western Arctic Ocean[52].

### Open water anomalies and SST baseline departure
Environmental data from the Beaufort Sea including SST (°C) and open water area (km²) data were retrieved from the NOAA OI SST v.2 (ref. 53) and the National Snow and Ice Data Center (NSIDC)[36] databases, respectively, for years when whales were harvested for subsistence purposes (2004–2022) and to compare with environmental baselines (1982–2011) (SST data, Supplementary Table 5; open water data, Supplementary Table 6).

July SSTs departures from baseline (that is, *z*-score correction) in the Beaufort Sea were calculated using the equation: July SST (°C) − mean July baseline (1982–2011) SST (°C))/standard deviation July baseline SST (°C) (Supplementary Table 5). Anomalies of open water in the Beaufort Sea were calculated by dividing summer monthly (June, July, August and September) open water areas (km²) by their respective monthly mean baseline (1982–2011) values (Supplementary Table 6).

Years were categorized into two DA prevalence groups; years with 100% of whales testing positive (more than 0 ng g⁻¹) for DA ($n = 7$ (years) with $n = 56$ total whales, '100% DA prevalence') and years with less than 100% DA prevalence ($n = 12$ (years) with $n = 149$ total whales, 'less than 100% DA prevalence') and compared to open water anomalies in the Beaufort Sea during summer months (June, July, August and September). Linear models were constructed to test whether the open water anomalies of each summer month (June–September) were significantly different during years of 100% DA prevalence and less than 100% DA prevalence in whales while weighting each model by the number of whales tested for DA per year. Weighted estimated marginal means of monthly open water anomalies for each DA prevalence group were tested using unpaired *t*-tests (Fig. 4a). All months (June, July, August and September) had significantly higher open water anomalies in the Beaufort Sea in years when DA prevalence was 100% compared to years when DA was present in less than 100% of whales harvested from the Beaufort Sea (Fig. 4a and Supplementary Table 3). Weighted estimated marginal mean comparisons were repeated for two more prevalence categories: (1) more than 90% versus less than 90% prevalence and (2) more than 75% versus less than 75% (Extended Data Fig. 2 and Supplementary Table 7). Although similar statistical relationships exist for the 90% prevalence comparison for all months, except September, they are not as significant as for the 100% prevalence comparisons (Fig. 4a, Extended Data Fig. 2a and Supplementary Tables 3 and 7a). For the greater than 75% prevalence comparison, the relationship remains for June, July and August as for 100% prevalence comparisons, but anomalies are similar among groups in September (Extended Data Fig. 2b and Supplementary Table 7b). For STX, all months (June, July, August and September) had significantly higher open water anomalies in the Beaufort Sea in years when STX prevalence was 100% compared to years when STX was present in less than 100% of whales harvested from the Beaufort Sea (Extended Data Fig. 3 and Supplementary Table 8).

Pearson correlations were performed among June open water anomalies and July SST anomalies for years associated with bowhead toxin analyses ($n = 19$ years) (Fig. 4b). Analysis was done using software

programs R[54] and R studio[55] and R packages emmeans[56], lme4 (ref. 57) and ggpubr[58].

## Average SSTs (1900–2023) for the different Seas

Average SST (°C) during May–September for years 1900–2023 in the Bering, Chukchi and Beaufort Seas (Fig. 5a–c) were obtained from the NOAA Extended Reconstructed SST v.5 data provided by the NOAA PSL, Boulder, Colorado, USA, from their website at https://psl.noaa.gov (ref. 35).

## Annual minimum sea ice (1979–2024) for the Seas

Annual minimum sea ice extent (km[2]) data for the Bering, Chukchi and Beaufort Seas during 1979–2024 were acquired from the NSIDC[36]. Daily sea ice extent for the Bering, Chukchi and Beaufort Seas were summed and the minimum daily extent for each year was plotted in Fig. 5d.

## Inclusion and ethics statement

This study was a mutually beneficial collaboration between NOAA, NWFSC, WARRN-West and the whaling communities of the NSB. The stakeholders' needs and concerns were the top priority for all aspects of this work. The project stemmed from a 15-year collaboration in service to the NSB Department of Wildlife Management (DWM), the Alaska Eskimo Whaling Commission and Whaling Captains' Associations, for health assessments of harvested bowhead whales.

## Reporting summary

Further information on research design is available in the Nature Portfolio Reporting Summary linked to this article.

## Data availability

Bowhead whale faecal algal toxin concentrations (DA and STX) and whale collection dates are available in Supplementary Table 1. *A. catenella* cyst data for 2018–2020 can be found at the Arctic Data Center database (https://doi.org/10.18739/A2HH6C70Z, http://doi.org/10.18739/A2833N123)[22,23]. *Alexandrium* cell density data are included in Supplementary Table 3. Hydrographic and velocity data from the mooring near Barrow Canyon were retrieved from the Arctic Observing Network Data Center and the DOI links for 2002–2022 are given in Supplementary Table 9 (refs. 49,50). Wind velocity and SLP data were provided by the European Centre for Medium-Range Weather Forecasts ERA5 reanalysis dataset (https://cds.climate.copernicus.eu/datasets/reanalysis-era5-single-levels)[51]. July SST data and open water area data during the summer months for the Beaufort Sea are provided in Supplementary Tables 4 and 5, respectively. SST data from 1900–2023 for the Bering, Chukchi and Beaufort Seas were obtained from the NOAA Extended Reconstructed SST v.5 data provided by the NOAA PSL, Boulder, Colorado, USA, from this link https://psl.noaa.gov/data/gridded/data.noaa.ersst.v5.html (ref. 35). The sea ice extent data for Bering, Chukchi and Beaufort Seas (1979–2024) were acquired from the NSIDC at the following link https://noaadata.apps.nsidc.org/NOAA/G02135/seaice_analysis/N_Sea_Ice_Index_Regional_Daily_Data_G02135_v3.0.xlsx (ref. 36).

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

**Acknowledgements** We thank the Alaska Eskimo Whaling Commission and the Barrow, Kaktovik and Nuiqsut Whaling Captains' Associations for granting access to their landed whales and for sharing traditional knowledge making this study possible. We acknowledge the many NSB DWM staff members over the years who assisted during bowhead whale harvest sampling. Bowhead faecal samples were collected under National Marine Fisheries Service permit no. 17350-00 issued to the NSB DWM. Research funding included: NOAA National Centers for Coastal Ocean Science ECOHAB grant no. NA20NOS4780195 (K.A.L. and D.M.A.), NIEHS R01 ES021930 (K.A.L.) and P01-ES028938-01 (D.M.A.), NSF R01s OCE-1314088 (K.A.L.) and OCE-1840381 (D.M.A.), NSF Office of Polar Programs grant no. OPP-1823002 (D.M.A.) and National Natural Science Foundation of China grant nos 42306251 and 42476262, National Key R&D Program of China 2024YFC2813202 and the Fundamental Research Funds for the Central Universities (P.L.). This is ECOHAB publication no. 1130. The scientific results, conclusions and opinions expressed herein are those of the authors and do not necessarily reflect the views of NOAA or the Department of Commerce.

**Author contributions** Conceptualization: K.A.L. and R.S. Methodology: K.A.L., R.S., P.C., P.L., R.S.P., K.A.H., B.D.B., G.S., E.K.B., D.M.A., E.F. and R.T. Investigation: K.A.L., R.S., P.C., P.L., R.S.P., K.A.H., G.S., D.M.A. and E.F. Visualization: K.A.L., P.C., P.L., R.S.P., B.D.B., G.S., E.F. and R.T. Funding acquisition: K.A.L., R.S., D.M.A. and R.S.P. Project administration: K.A.L. and R.S. Supervision: K.A.L., R.S. and G.S. Writing—original draft: K.A.L., R.S., P.C., P.L., R.S.P., K.A.H., B.D.B., G.S., E.K.B., D.M.A., E.F. and R.T. Writing—review and editing: K.A.L., R.S., P.C., P.L., R.S.P., K.A.H., B.D.B., G.S., E.K.B., D.M.A., E.F. and R.T.

**Competing interests** The authors declare no competing interests.

**Additional information**
**Correspondence and requests for materials** should be addressed to Kathi A. Lefebvre.

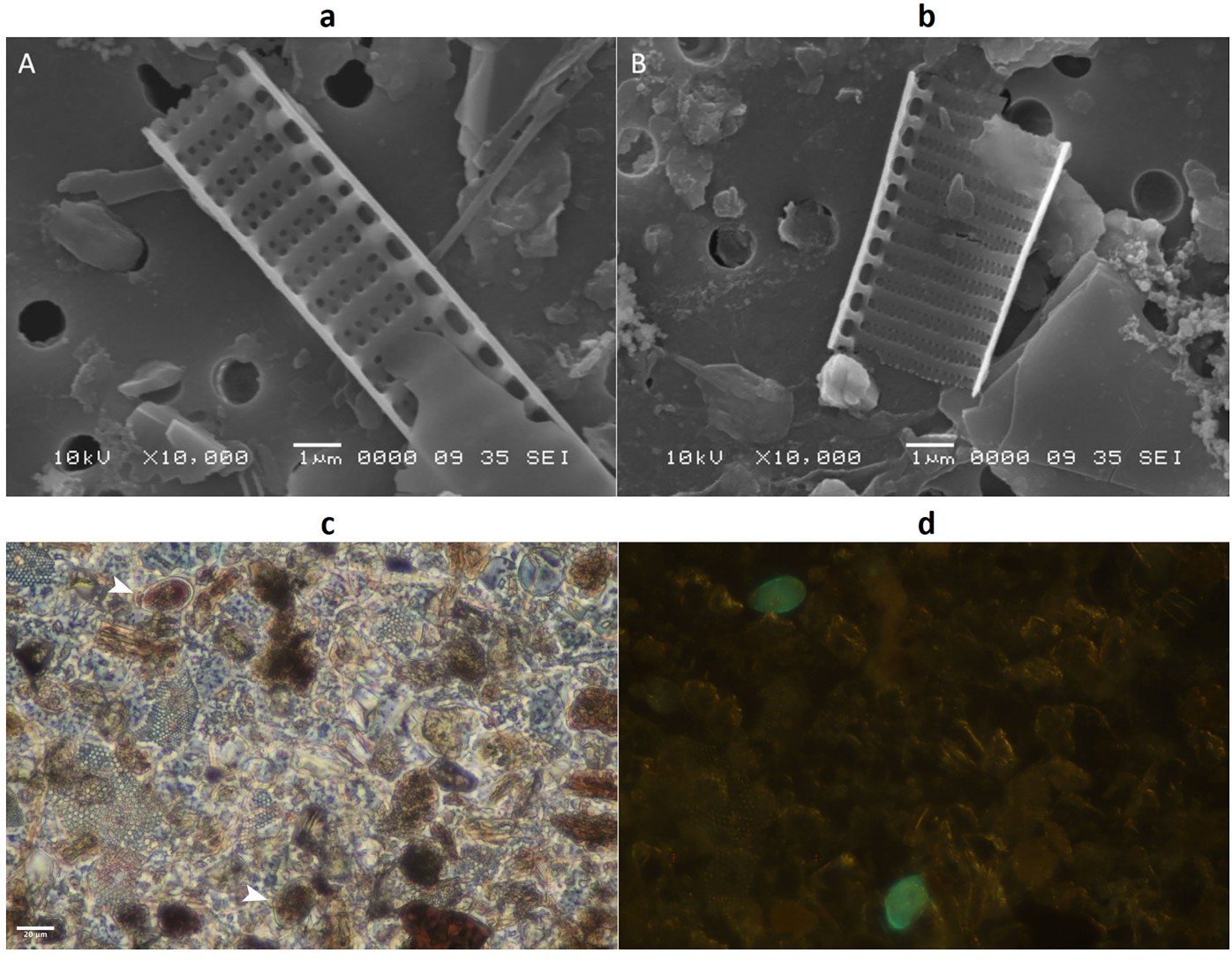

**Extended Data Fig. 1 | Scanning electron microscope (SEM) and fluorescence microscope images. a**, partial *Pseudo-nitzschia* cf. *pungens* frustule and **b**, partial *Pseudo-nitzschia* cf. *seriata* frustule isolated from bowhead whale fecal samples that contained domoic acid. **c**, sediment sample containing two *Alexandrium* cysts (white arrows) viewed with normal phase contrast light microscopy and **d**, the same sediment sample stained with primuline and viewed with fluorescence. White scale bars = 1 μm in panels a - b, and 20 μm in panels c and d.

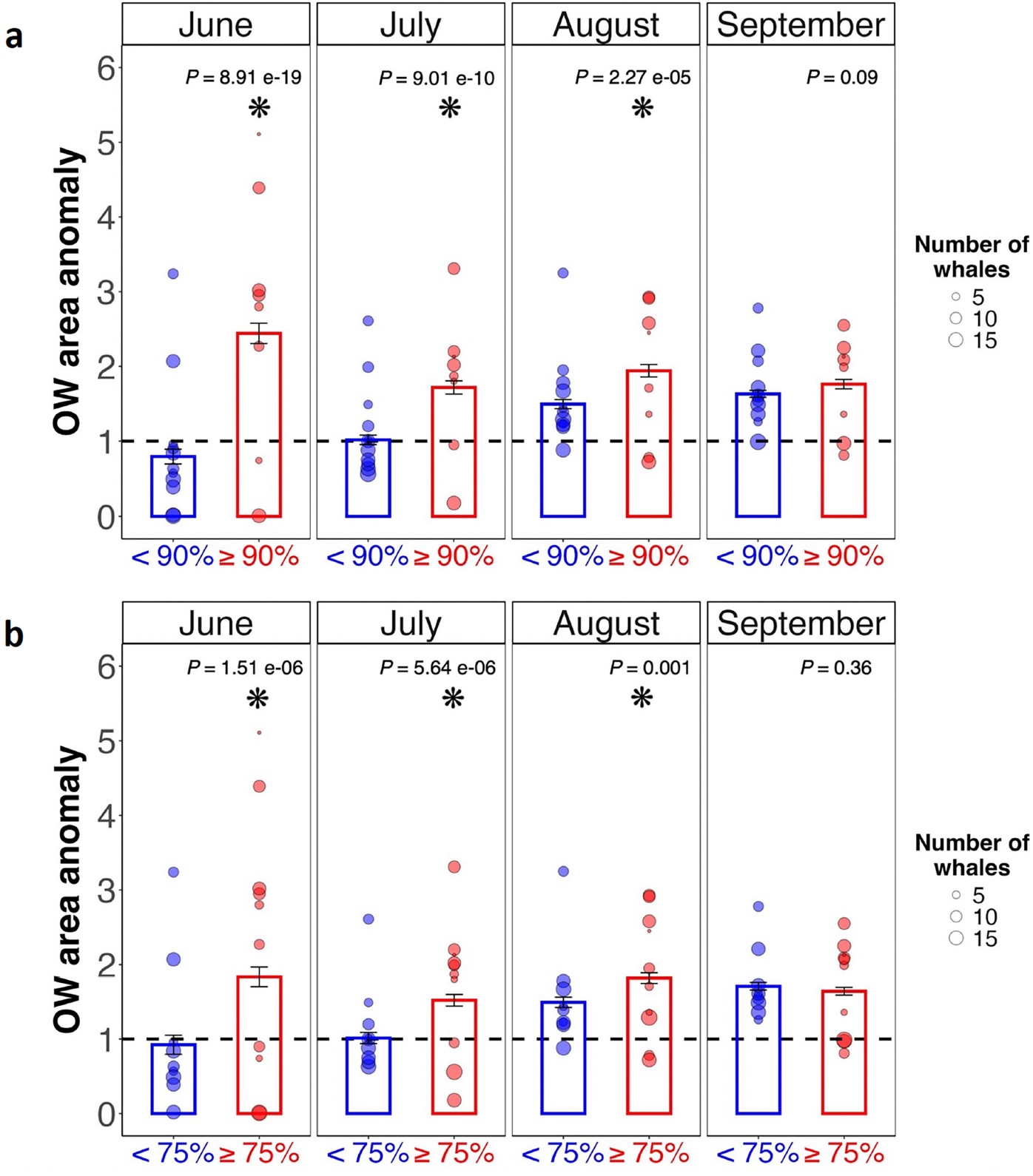

**Extended Data Fig. 2 | Open water (OW) area anomalies with domoic acid (DA) prevalence.** Comparison of estimated marginal means (EMM ± standard error) of Beaufort Sea OW area anomalies in June, July, August and September between yearly whale DA prevalence groups; (**a**) < 90% (blue bars and points; n = 134 whales sampled across Y = 11 independently sampled years) and ≥ 90% (red bars and points; n = 71 whales sampled across Y = 8 independently sampled years) and (**b**) < 75% (blue bars and points; n = 106 whales sampled across Y = 9 independently sampled years) and ≥75% (red bars and points; n = 99 whales sampled across Y = 10 independently sampled years) domoic acid (DA). Size of points represent the total whales sampled for that year (n = 3–19 whales per year). OW anomalies are calculated by dividing monthly OW areas (km²) by the monthly baseline areas (km²), resulting in a unitless anomaly equating baseline OW area values to 1 (dashed black line). * = statistically significant difference of EMM OW anomaly values between DA prevalence groups (unpaired two-sided t-tests with no P-value adjustments; Table S7).

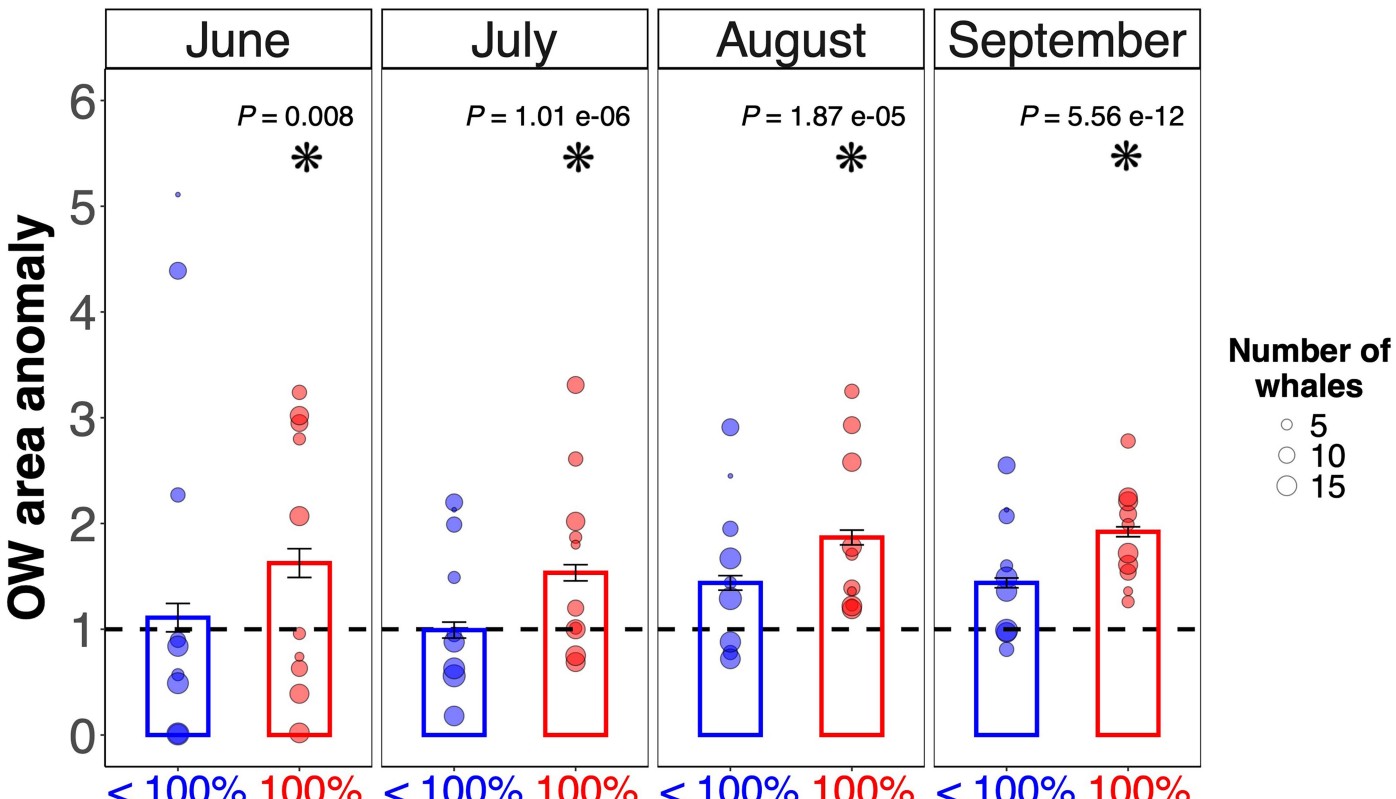

**Extended Data Fig. 3 | Open water (OW) area anomalies with saxitoxin (STX) prevalence.** Comparison of estimated marginal means (EMM ± standard error) of Beaufort Sea OW area anomalies in June, July, August and September between two yearly prevalence groups; (1) < 100% (blue bars and points; n = 104 whales sampled across Y = 9 independently sampled years) and (2) 100 % (red bars and points; n = 101 whales sampled across Y = 10 years) STX. Size of points represent the total whales sampled for that year (n = 3–19 whales per year). OW anomalies are calculated by dividing monthly OW areas (km²) by the monthly baseline areas (km²), resulting in a unitless anomaly equating baseline OW area values to 1 (dashed black line). * = statistically significant difference of EMM OW anomaly values between STX prevalence groups (unpaired two-sided t-tests with no P-value adjustments; Table S8).

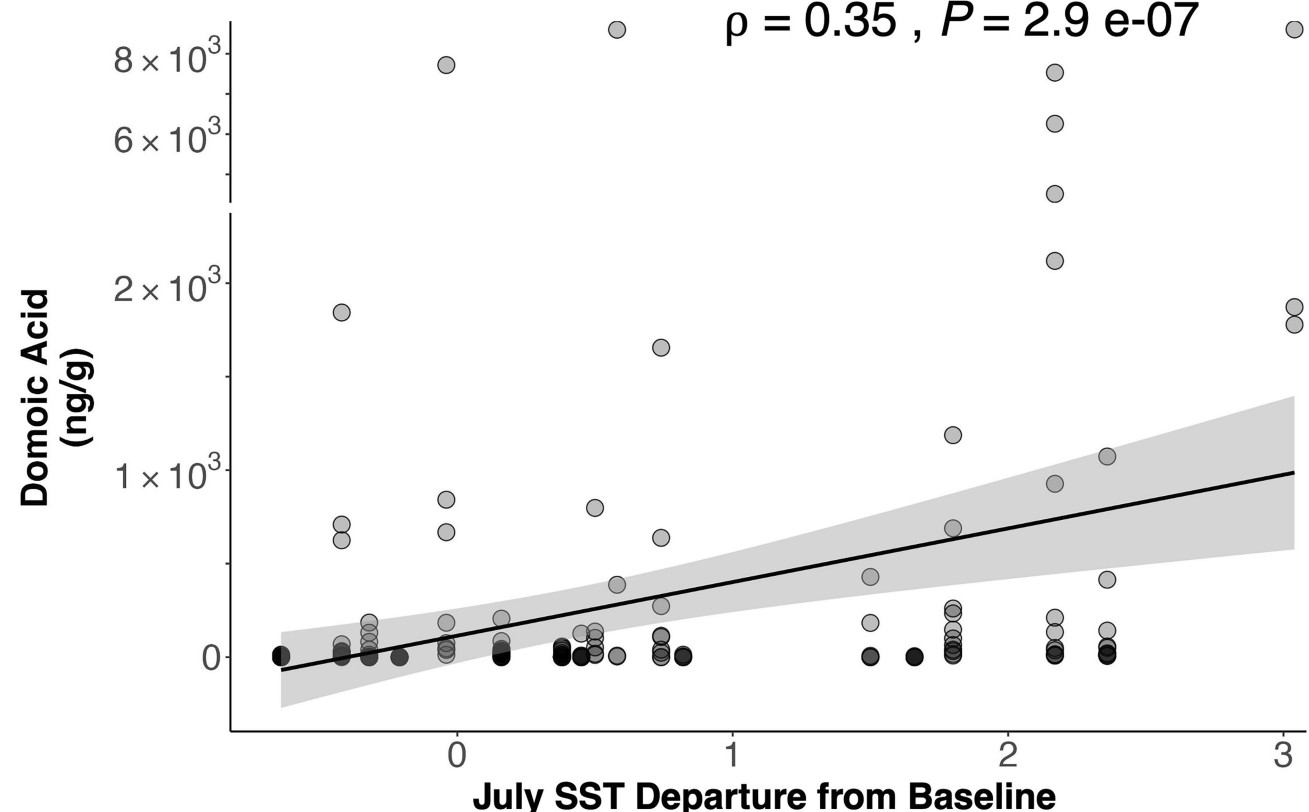

**Extended Data Fig. 4 | Sea surface temperature (SST) anomalies with domoic acid (DA) concentrations.** Spearman rank correlation results comparing domoic acid (DA) concentrations in bowhead feces (ng/g; n = 205) with July sea surface temperature (SST) anomalies. DA concentrations in feces were the most strongly correlated with July summer SST anomalies (rho = 0.35, $P$ = 2.9 e-07, Figure below). DA was also positively and significantly correlated with June (rho = 0.17, $P$ = 0.016), August (rho=0.28, $P$ = 6.2 e-05), and September (rho = 0.25, $P$ = 2.7 e-4) SST anomalies. Saxitoxin (STX) concentrations in bowhead feces were not significantly correlated with SST anomalies (June; rho = 0.13, $P$ = 0.071, July; rho = 0.11, $P$ = 0.13, August; rho = 0.08, $P$ = 0.24, and September; rho = 0.04, $P$ = 0.58).

# Reporting Summary

Please do not complete any field with "not applicable" or n/a.  Refer to the help text for what text to use if an item is not relevant to your study.
For final submission: please carefully check your responses for accuracy; you will not be able to make changes later.

## Statistics

For all statistical analyses, confirm that the following items are present in the figure legend, table legend, main text, or Methods section.

| n/a | Confirmed | |
|---|---|---|
| ☐ | ☒ | The exact sample size (*n*) for each experimental group/condition, given as a discrete number and unit of measurement |
| ☐ | ☒ | A statement on whether measurements were taken from distinct samples or whether the same sample was measured repeatedly |
| ☐ | ☒ | The statistical test(s) used AND whether they are one- or two-sided *Only common tests should be described solely by name; describe more complex techniques in the Methods section.* |
| ☐ | ☒ | A description of all covariates tested |
| ☐ | ☒ | A description of any assumptions or corrections, such as tests of normality and adjustment for multiple comparisons |
| ☐ | ☒ | A full description of the statistical parameters including central tendency (e.g. means) or other basic estimates (e.g. regression coefficient) AND variation (e.g. standard deviation) or associated estimates of uncertainty (e.g. confidence intervals) |
| ☐ | ☒ | For null hypothesis testing, the test statistic (e.g. *F*, *t*, *r*) with confidence intervals, effect sizes, degrees of freedom and *P* value noted *Give P values as exact values whenever suitable.* |
| ☒ | ☐ | For Bayesian analysis, information on the choice of priors and Markov chain Monte Carlo settings |
| ☒ | ☐ | For hierarchical and complex designs, identification of the appropriate level for tests and full reporting of outcomes |
| ☐ | ☒ | Estimates of effect sizes (e.g. Cohen's *d*, Pearson's *r*), indicating how they were calculated |

*Our web collection on statistics for biologists contains articles on many of the points above.*

## Software and code

Policy information about availability of computer code

| Data collection | No software was used for data collection. |
|---|---|
| Data analysis | Comparisons of environmental data (open water area anomalies) and algal toxin prevalence groupings of whales was done using freely available computer software. Analysis was done using software programs R (version 4.4.2) and R studio (version 2024.09.1+394). Specifically we used the following R packages: "lme4" (v. 1.1-25.5) for constructing linear models, estimated marginal means were generated and compared using pairwise comparisons (unpaired t-tests) among whale groupings using "emmeans" (v. 1.10.5), and Pearson correlation analysis of June open water anomalies and July sea surface temperature anomalies was done using the package "ggpubr" (v. 0.6.0). The calculations of heat flux and wind/sea level pressure composites were done using MATLAB 2024b.  All software programs and packages are properly cited in the manuscript. |

For manuscripts utilizing custom algorithms or software that are central to the research but not yet described in published literature, software must be made available to editors and reviewers. We strongly encourage code deposition in a community repository (e.g. GitHub). See the Nature Portfolio guidelines for submitting code & software for further information.

## Data

Policy information about availability of data

All manuscripts must include a data availability statement. This statement should provide the following information, where applicable:

- Accession codes, unique identifiers, or web links for publicly available datasets
- A description of any restrictions on data availability
- For clinical datasets or third party data, please ensure that the statement adheres to our policy

Bowhead whale fecal algal toxin concentrations (DA and STX) and whale collection dates are available in the supplementary materials (Table S1). Alexandrium catenella cyst data for 2018 – 2020 can be found at the Arctic Data Center database (https://doi:10.18739/A2RF5KG8J , https://doi:10.18739/A2Q814V0P )22. Alexandrium cell density data are included in the supplementary materials (Table S3). Hydrographic and velocity data from the mooring near Barrow Canyon were retrieved from the Arctic Observing Network Data Center and the DOI links for 2002-2022 are given in Supplementary Table S948,49. Wind velocity and sea level pressure data were provided by the European Centre for Medium-Range Weather Forecasts (ECMWF) ERA5 reanalysis dataset (https://cds.climate.copernicus.eu/datasets/reanalysis-era5-single-levels )50. July sea surface temperature (SST) data and open water area data during the summer months for the Beaufort Sea are provided in the supplementary material (Table S4 and Table S5, respectively). SST data from 1900 – 2023 for the Bering, Chukchi, and Beaufort Seas were obtained from the NOAA Extended Reconstructed SST V5 data provided by the NOAA PSL, Boulder, Colorado, USA, from this link https://psl.noaa.gov/data/gridded/data.noaa.ersst.v5.html 34. The sea ice extent data for Bering, Chukchi, and Beaufort Seas (1979 – 2024) were acquired from the National Snow and Ice Data Center at the following link https://noaadata.apps.nsidc.org/NOAA/G02135/seaice_analysis/N_Sea_Ice_Index_Regional_Daily_Data_G02135_v3.0.xlsx35.

## Research involving human participants, their data, or biological material

Policy information about studies with human participants or human data. See also policy information about sex, gender (identity/presentation), and sexual orientation and race, ethnicity and racism.

| | |
|---|---|
| Reporting on sex and gender | No human samples |
| Reporting on race, ethnicity, or other socially relevant groupings | No human samples |
| Population characteristics | No human samples |
| Recruitment | No human samples |
| Ethics oversight | No human samples |

Note that full information on the approval of the study protocol must also be provided in the manuscript.

# Field-specific reporting

Please select the one below that is the best fit for your research. If you are not sure, read the appropriate sections before making your selection.

☐ Life sciences  ☐ Behavioural & social sciences  ☒ Ecological, evolutionary & environmental sciences

For a reference copy of the document with all sections, see nature.com/documents/nr-reporting-summary-flat.pdf

# Ecological, evolutionary & environmental sciences study design

All studies must disclose on these points even when the disclosure is negative.

| | |
|---|---|
| Study description | Bowhead fecal samples were collected from landed whales during indigenous fall harvest seasons over 19 years in order to measure the presence of algal toxins in Arctic food webs. |
| Research sample | Feces extracted from the bowel of harvested whales. |
| Sampling strategy | All samples used in this study were comparable in terms of collection techniques and time of year (Fall harvest season). |
| Data collection | Fecal samples were analyzed for the presence of the algal toxins saxitoxin and domoic acid. |
| Timing and spatial scale | Bowhead whales were havested from the Beaufort Sea from August to October from 2004 to 2022. |
| Data exclusions | Not applicable |
| Reproducibility | Not applicable |
| Randomization | Not applicable |

| Blinding | Not applicable |
|---|---|

**Did the study involve field work?**  ☒ Yes  ☐ No

## Field work, collection and transport

| Field conditions | Arctic conditions over 19 years of whale harvesting. |
|---|---|
| Location | Whales were landed at Utqiaġvik, Alaska. |
| Access & import/export | All samples were collected in collaboration with the North Slope Borough Department of Wildlife Management as part of their whale health monitoring program. |
| Disturbance | Traditional hunting practices were used for harvesting whales. Fecal samples were collected post mortem. |

# Reporting for specific materials, systems and methods

We require information from authors about some types of materials, experimental systems and methods used in many studies. Here, indicate whether each material, system or method listed is relevant to your study. If you are not sure if a list item applies to your research, read the appropriate section before selecting a response.

### Materials & experimental systems

| n/a | Involved in the study |
|---|---|
| ☒ | ☐ Antibodies |
| ☒ | ☐ Eukaryotic cell lines |
| ☒ | ☐ Palaeontology and archaeology |
| ☐ | ☒ Animals and other organisms |
| ☒ | ☐ Clinical data |
| ☒ | ☐ Dual use research of concern |
| ☒ | ☐ Plants |

### Methods

| n/a | Involved in the study |
|---|---|
| ☒ | ☐ ChIP-seq |
| ☒ | ☐ Flow cytometry |
| ☒ | ☐ MRI-based neuroimaging |

## Animals and other research organisms

Policy information about studies involving animals; ARRIVE guidelines recommended for reporting animal research, and Sex and Gender in Research

| Laboratory animals | Laboratory animals were not used in this study. |
|---|---|
| Wild animals | No live wild animals were used in this study. Fecal samples were collected opportunistically post mortem from bowhead whales harvested for subsistence uses by Native tribal communities. |
| Reporting on sex | Sex was not reported. |
| Field-collected samples | Bowhead whale fecal samples were collected post mortem during harvest processing of whales by indigenous hunters and North Slope Borough Department of Wildlife Management staff. |
| Ethics oversight | Ethics oversight was not needed. Fecal samples were collected from deceased whales under National Marine Fisheries permit number 17350-00 |

Note that full information on the approval of the study protocol must also be provided in the manuscript.

## Plants

| | |
|---|---|
| Seed stocks | NA |
| Novel plant genotypes | NA |
| Authentication | NA |

