## [Peer Review File · Nature]

Bowhead whale feces link increasing algal toxins in the Arctic to ocean warming

Corresponding Author: Dr Kathi Lefebvre

Version 0:

Reviewer comments:

Referee #1

(Remarks to the Author)

In an emerging field of ecosystem sentinel science, this paper is a stand-out. I have recently co-authored two scoping reviews on the topic (Hazen et al. 2019, 2024), and this study rises above most of what has been published to date. The mechanistic links between physical environment, primary producers, and top predators are made so clearly, with large effect sizes, multiple lines of evidence, over decades of data collection. As a result this paper will be broadly interesting to those studying the effects of climate change on the oceans, marine mammals, harmful algal blooms, and physical/chemical/biological oceanography to name a few. The figures are well made, contain the appropriate information, and are visually appealing (though legibility would be improved on some, e.g., larger text for the axes in Fig. 5). I do not have the necessary expertise to evaluate the details of the oceanographic modeling and algal data extraction and analysis; so did not thoroughly review those methods; that said, nothing jumps out to me as problematic. All this said, I do have a few major comments below that would need to be well addressed before I can endorse the paper's publication in Nature.

Major comments:

The work feels like parachute science to me. Of the 12 authors of the paper, only three are based in Alaska, and it seems like none are members of the native Alaskan community that were essential in the collection of the key samples (bowhead whale fecal samples) that would have been impossible to acquire otherwise. Why is that? I could be wrong and may be overlooking a Native American co-author, and please let me know if this is the case. I understand that co-authorships should not be awarded without merit, but I can only imagine the amount of work native harvesters had to do to make this paper possible. This deserves more than one sentence of credit in the Acknowledgments section as is currently the case. Please provide an explanation for this omission.

As I mentioned in my overview of the paper above, this work has the potential to be a paragon of what an ecosystem sentinel study can be. However, the ecosystem sentinel literature is not cited at all. I find this omission confusing. In addition to the two review papers I mentioned above (Hazen et al. 2019, 2024), others you may consider drawing in and citing that are particularly relevant are (Scholin et al. 2000, Moore 2008, Bossart 2011, Bargu et al. 2012, Backer and Miller 2016, Plön et al. 2024). The lack of any of this literature being cited and discussed in the paper is glaring. Develop the ecosystem sentinel framework more in the main text as it's the integrative concept that ties this interdisciplinary study together. Topics you may consider discussing, are bowheads an example of a leading or elucidating sentinel (*sensu* Hazen et al. 2019)?; how would you recommend the knowledge gained in this study be used monitor for HABs in this region in the future? What does this work say about the future of ecosystem sentinel science? And so on.

There is excellent discussion and contextualization of the relevant oceanography and the algal species' ecology in the paper. However, this is mirrored by a near-complete lack of the ecology of bowhead whales and calanoid copepods (and other small zooplankton), the whales' main prey. Why is this? It is never even stated that bowhead whales are there to feeding on zooplankton, and presumably ingest phytoplankton incidentally (or contained within their copepod prey?). In a general journal like Nature, you need to be more explicit about the basic ecology of your study species and cannot assume the reader will know this. All this context, with supporting references, is sorely needed.

I am curious how the low, medium, and high thresholds for DA and STX were chosen – was it from empirical cutoffs in the raw data, known risk thresholds to wildlife, something else? Why not just keep these quantitative concentration values as a

continuous variable? More explanation is needed to justify this approach.

Minor comments:

Line 48-49: remove vague language like “since time immemorial” with real information (e.g., “since at least 12000 BCE” or whatever is accurate) here and elsewhere in the paper.

Fig 4B: Add error band, R2 value, or some way for the reader to easily evaluate the fit of this relationship.

Fig. 5: Make axes labels larger, hard to see now as they are too small.

Literature cited

Backer, L. C., and M. Miller. 2016. Sentinel Animals in a One Health Approach to Harmful Cyanobacterial and Algal Blooms. *Veterinary Sciences* 3:8.

Bargu, S., M. W. Silver, M. D. Ohman, C. R. Benitez-Nelson, and D. L. Garrison. 2012. Mystery behind Hitchcock's birds. *Nature Geoscience* 5:2–3.

Bossart, G. D. 2011. Marine mammals as sentinel species for oceans and human health. *Veterinary Pathology* 48:676–690.

Hazen, E. L., B. Abrahms, S. Brodie, G. Carroll, M. G. Jacox, M. S. Savoca, K. L. Scales, W. J. Sydeman, and S. J. Bograd. 2019. Marine top predators as climate and ecosystem sentinels. *Frontiers in Ecology and the Environment* 17:565–574.

Hazen, E. L., M. S. Savoca, T. J. Clark-Wolf, M. Czapanskiy, P. M. Rabinowitz, and B. Abrahms. 2024. Ecosystem Sentinels as Early-Warning Indicators in the Anthropocene. *Annual Review of Environment and Resources* 49:573–598.

Moore, S. E. 2008. Marine mammals as ecosystem sentinels. *Journal of Mammalogy* 89:534–540.

Plön, S., K. Andra, L. Auditore, C. Gegout, P. J. Hale, O. Hampe, M. Ramilo-Henry, P. Burkhardt-Holm, A. M. Jaigirdar, L. Klein, M. K. Maewashe, J. Müssig, N. Ramsarup, N. Roussouw, R. Sabin, T. C. Shongwe, and P. Tuddenham. 2024. Marine mammals as indicators of Anthropocene Ocean Health. *npj Biodiversity* 3:1–9.

Scholin, C. A., F. Gulland, G. J. Doucette, S. Benson, M. Busman, F. P. Chavez, J. Cordaro, R. DeLong, A. De Vogelaere, J. Harvey, M. Haulena, K. Lefebvre, T. Lipscomb, S. Loscutoff, L. J. Lowenstine, R. Marin, P. E. Miller, W. A. McLellan, P. D. R. Moeller, C. L. Powell, T. Rowles, P. Silvagni, M. Silver, T. Spraker, V. Trainer, and F. M. Van Dolah. 2000. Mortality of sea lions along the central California coast linked to a toxic diatom bloom. *Nature* 403:80–84.

Referee #2

(Remarks to the Author)

Bowhead whale feces link increasing algal toxins in the Arctic to ocean warming

Kathi A. Lefebvre et al.

This study links the rapid onset of climate change in the Arctic Ocean to the expansion of harmful algal blooms (HABs) and subsequent transfer of algal toxins to planktivorous bow whales in this system near Alaska. The study is very highly interdisciplinary and brings together biologists, chemists, physical oceanographers, atmospheric scientists, and indigenous peoples who are engaged in long-term monitoring. Beyond its impressive breadth, the study also has a remarkably unique data set and reveals intriguing linkages between changing climate, HABs, and trophic transfer of toxins. And while there is some good evidence to support these linkages, there are also outstanding questions, of course.

The Arctic Ocean is a large ecosystem. The mooring used to track temperatures, velocities, and heat flux is located at a fixed point 150 km east of Pt. Barrow. There was no indication of where the 205 whales were collected from for this study relative to the mooring site. Placing points on a map would be very helpful. The authors argue, these animals are integrating the water column, so detailing this information will help make this case, to some extent, by showing how far afield the trends persisted.

Arguably, the most important figures in the paper is Figure 2b and 2c. Are there any statistics that can be applied to Figure 2b and 2c to identify the precise period when the groups are significantly different from each other instead of simply calling out the peak date?

Figure 2b and 2c. Why is the heat flux so much larger for SXT compared to DA?

Figure 2b and 2c. Was there a reason for the specific bin sizes used?

Whale feeding: What size particles do the whales feed on, precisely? If they consume copepods, might they directly consume concatenated HAB cells that could be on a similar size. Also, as what depths do they feed? How does this compare to where HABs occur?

Could toxins have degraded in the stored samples over this 20-year study? Is there a control for this? Given the temperatures have been rising over time (higher toxins prevalence in recent samples) and that older samples would be more likely to have degraded toxins (higher toxins prevalence in recent samples), does degradation of toxins need to be considered? Can it be discounted?

The ELISA kits were used to measure toxins. The SXT ELISA has the advantage of reacting with all congeners, but its efficacy for doing so varies with each congener. How does its specificity for specific congeners match up with the congeners most prevalent in the strains of *A. catenella* prevalent in the Arctic? Does this need to be considered?

Which direction was the heat flux? I see the value '125°T' but is that vertical, horizontal? N, S, E, W? I think this should be stated outright and should be discussed with regard to the relevance for the development of each HAB.

Open water area were linked to DA prevalence. There is no information of how this related to SXT.

It is important to state whether toxin levels were correlated with just surface water temperatures; it appears they were not which is probably because the heat fluxes are more important but this is a case to further support this study.

Differences between SXT and DA. There was no investigation of OW area and SXT and one would presume this is because it did not exist. Similarly, while the differences in prior heat flux among the binned toxin levels were clear and large for DA, they were smaller and more muddled for SXT. Together, this suggests that temperature changes and OW changes are more important for DA; why might that be?

This study is very focused on the HABs and toxins in feces, but an important step between these two outcomes is the consumption of the HABs by zooplankton which are actually consumed by the whales. This merits consideration and discussion. How might changes in the relationship between the HABs and zooplankton have affected by temperatures and, in turn, affected the outcomes of this study?

Much of the prior work describing the dynamics of Alexandrium blooms, cells, and cysts and seeding of the (by D. M. Anderson and others) seem to have been focused on regions south of the Bering Strait and accounting for how blooms in that region seed the Chukchi Sea and, to lesser extent how the Chukchi Sea might seed the southern Beaufort Sea. Given this, it would seem a broadening of the implications of this study would have value. Might the same trends seen here hold for the Bering Sea? The (southern) Chukchi Sea? It would be nice to see the explanation of Figure 3 broadened to include outcomes for the whole region.

Line 35: 'Negative' instead of 'Devastating'; less drama

Line 47: 'supplies' instead of 'security';

Line 48: For 'millennia' instead of 'since time immemorial'; accurate quantification.

Lines 64-67: Consider combining sentences and clarifying what was monitored and precisely why.

Line 69: Is their diet exclusively zooplankton? What is known regarding their minimum particle size retention? Might they directly consume concatenated HAB cells?

Line 82: Increased risks or increase occurrence?

Lines 85-88: This statement should be supported by references to publications

Lines 110-112: Seems like there should be references to cite for new genotype /populations of *Pseudo-nitzschia* that was 'previously recorded'; cite this previous record.

Lines 110-113: Do *P. seriata*, *P. pungens*, and *P. obtusa* all make DA? All at high levels? Details with references here are needed.

Lines 114-119: Are these hypotheses necessary? Seems clear

Figure 1a: What does the horizontal distribution of the data represent in the plot?

Figure 1a is a little small and unclear; red and green dashed lines are faint, hard to see.

Fig 1C. Indicate in Fig 1b where Fig1C is located.

Line 145: Velocity of what and in what direction? Horizontal? Vertical? More detail needed. Also, doesn't the direction of the velocity matter a lot with regard to stated hypotheses?

Line 149: Did the heat flux affect HAB toxin levels in bowhead whales or did it effect the HABs and food web/zooplankton?

Line 152: Why 20 days? Why not more or less.

Line 156: "with the largest effect beyond 13 days prior"...The meaning of this statement is unclear.

Line 161: "within 10 days prior, the high STX group (>50 ng/g) had the highest heat flux". Mis-stated. The STX did not have a heat flux.

Line 163-174: Might not the difference in heat flux time and HAB be longer for Alexandrium given it starts as a benthic cysts in sediments? Are these heat fluxes important for cysts?

Lines 204-204: "which were progressively weaker corresponding to times of fecal collections that contained higher toxin concentrations". Awkward phrasing.

Figure 3: Why are the regions considered for sea level pressure and wind velocity different? It would be useful to show the area of focus show in a and b in the plots for c and d.

The correlation between OW and temperature (Fig 4b) is not directly related to this study.

Open water area linked to DA prevalence: There needs to be a statement of how OW related to SXT

Line 266-307: Yes, the region is warming, but is this directly related to the toxins in the feces? Is there any direct correlation between the toxin in feces and temperature? Figure 5 is interesting, but seems like a distraction from the data being presented in the paper. Is absolute temperature important or heat flux? If it is the heat flux, rather than absolute temperature, this provides a more nuanced story that gets eroded by simple plots of rising temperatures.

Referee #3

(Remarks to the Author)

This manuscript reports on innovative, compelling, and important research linking Arctic climate and ocean conditions to the presence and intensity of toxic algae found in fecal samples obtained from bowhead whales that were taken during aboriginal subsistence harvests from 2004-2022. Key results reported here show that a combination of environmental conditions that include warm ocean temperatures, large ice-free open water area, weak NE winds along Alaska's NW coast, and increased heat flux to toxic algae concentrations in whale fecal samples. Trends in open water and ocean temperature provide a link to Arctic warming linked with global warming. This work is likely to generate broad interest from scientists, resource managers, aboriginal communities in the Arctic, and the general public. The article is very well written and the methods, results, and discussion points should be easy to understand for a broad audience save for one relatively minor issue that I flagged regarding the description of method used to create Fig. 3. Overall I liked this manuscript a lot and recommend that it be accepted for publication pending minor revisions. I have a few general comments for the authors to consider below, followed by specific comments for given line numbers, figures or tables. I congratulate the authors on this engaging, timely, interesting and important contribution.

General Comments:

1. Were fecal samples collected from years prior to 2004 analyzed for presence of neurotoxins? If not, can they be analyzed to provide a longer historical context? It seems like samples from periods with consistently less open water/more extensive sea ice (like in the 1970s-early 1990s, shown in your Fig 5d) would be very valuable for documenting a shift to increased HAB activity in your study period that is characterized by mostly low sea-ice extent/high open-water area and warm SSTs (especially since 2007). That said, your study period does contain substantial inter annual variation in those factors, but that variation is around a lower-ice baseline.

2. This work has some parallels with a study of *A. catenella* blooms in Puget Sound that was done about 15 years ago. That work also searched for environmental factors associated with bloom events sampled over two decades, and identified a combination of atmospheric/hydrologic/oceanographic factors were at play. The approach also considered the daily evolution of these factors. One issue that the Puget Sound work brought to mind while reviewing your manuscript is the daily evolution of surface winds, and how the associated "weak NE wind events" may have varied over the course of your composite events and over recent years and decades. Have you tried, or considered, developing a daily NE wind event index, or a daily Beaufort High pressure index? A daily atmospheric forcing index could be based on SLP (180-120W, 70-80N), or an index of NE wind speed in the 69-71N, 170W-145W region. You could then (a) document the daily evolution of wind forcing associated with composite events (similar to what you've done with heat flux), and (b) summarize the annual frequency, intensity, and duration of "weak NE wind events" associated with toxic algae events from the fecal samples. I raise this issue because understanding the history of favorable atmospheric forcing will complement what you've done with the annual time series of open water area and SST, and it adds a critical piece of your mechanism needed to go back in time or forward in time with forecasts and future projections. The Puget Sound work is summarized by Moore et al (2009) and Moore et al. (2011). I realize that adding a deeper dive into the atmospheric forcing at this point is probably more than what you'd like to do with this manuscript, and I don't think it is needed to bolster what is currently reported. But it would be a nice

addition to what you have already done, and could be part of another manuscript.

Moore, S.K., N.J. Mantua, B.M. Hickey, and V.L. Trainer. 2009. Recent trends in paralytic shellfish toxins in Puget Sound, relationships to climate, and capacity for prediction of toxic events. *Harmful Algae*, doi:10.1016/j.hal.2008.1003.

Moore, S.K., N.J. Mantua, E.P. Salathé Jr. 2011. Past trends and future scenarios for environmental conditions favoring the accumulation of paralytic shellfish toxins in Puget Sound shellfish. *Harmful Algae*, In Press, Corrected Proof, Available online 15 April 2011, ISSN 1568-9883, DOI: 10.1016/j.hal.2011.04.004. <http://dx.doi.org/10.1016/j.hal.2011.04.004>

3. For Fig 3, it is not clear how the composite wind vector maps were created. Lines 200-1 state that “the wind composites correspond to 15 days and 10 days before each fecal sample was collected ...”. I take this to mean the maps depict daily-mean surface wind vectors at the stated time points. If these are single-day daily mean fields please specify that in the Figure caption. Related to my comment 2 above, what are the typical time-scales of weak NE wind events associated with high toxicity samples (a few days, a week)?

Line-specific comments:

64: Were fecal samples from years prior to 2004 analyzed for presence of neurotoxins? If not, why not? It seems like samples from periods with consistently less open water/more extensive sea ice (like in the 1970s-early 1990s, shown in your Fig 5d) would be very valuable for documenting a shift to increased HAB activity.

163-4: This analysis provides compelling evidence linking recent history of ocean heat flux to HAB toxin concentrations in bowhead whale feces, and that warmer ocean temperatures are linked to higher HAB toxin loads in the food web. Do these events also involve other aspects of HAB bloom dynamics that follow patterns of enrichment, stratification, nutrient uptake, nutrient stress etc seen in other systems?

186: indicate sub-sample sizes (number of events) used for the different sub-group composites; do Fig 2 panels (b) and (c) indicate a “relaxation” of strong advection events in the few days prior to sampling? I’m just trying to understand the time-evolution of the composite events

200-1: for the 10 and 15 day composites, are these multi-day windows centered on 15 and 10 days prior, or single-day daily mean fields?

235: again, are these composites of daily data for a single day or some window of days centered on 15 and 10 days prior to fecal sampling?

271, 820, 847, 874: are OW area anomalies in units of 100,000 sq km? Please indicate on the y-axis label or figure captions and in the top row of Tables S3 and S7 for EMM Anomaly

291-4: Are there also long-term trends in June-July-Aug atmospheric forcing patterns (the Beaufort High, or associated multi-day events with weak NE winds along the Beaufort coast), or have those been unchanged while trends in warming ocean temps and sea ice loss are driven by thermodynamic processes? You can probably come up with a daily index of atmospheric forcing based on your composite SLP, or an index of NE wind speed in the 69-71N, 170W-145W region.

Version 1:

Reviewer comments:

Referee #1

(Remarks to the Author)

I appreciate the care with which the authors responded to my comments on their paper; the paper is improved and my following suggestions are minor and should not preclude publication.

I realize that for many people, publication is not an important metric, nor one they care about. Thank you for explaining more about the collaboration between NWFSC, NSB-DWM, and the Inupiaq community. This is sufficient for me, but I wonder if you can have a supplemental section that can also explain this? It could be largely cut-and-paste from the response you provided with this revision. This would be helpful, if possible, because with a high-profile paper like this, I think people will want to know more about this network, and to perhaps be reassured that the relationship between all parties is collaborative and not exploitative.

My apologies for not noticing that you had cited the Scholin et al. (2000) paper. Including that and the Hazen et al (2019) paper is helpful, especially considering the limited number of references you’re afforded at Nature. And thank you for including some of the essential relevant ecology of the other species in this food chain.

Your replies to my other comments are suitable, thank you.

Nice work,
Matt Savoca

Referee #2

(Remarks to the Author)
I am pleased with the revision.

The following points should be added to the manuscript and specifically the differences in depths between the HABs and the depths where whales are feeding. Suggests migration by zooplankton helps transport toxins:
"Bowhead whales are known to feed near Utqiagvik at depths ranging from shallow continental shelf (45m) to the deeper waters (>300m) of Barrow Canyon (Citta et al. 2015)(Sheffield and George 2021). Depths at which Alexandrium cells mainly occur are in the upper 25m (Anderson et al. 2021) and Pseudo nitzschia particulate DA measurements have been documented from the surface (~2 m) to chl-a max depths (20 - 40m) (Hubbard et al. 2023)."

The following points should be added to the manuscript. Supplemental methods would be fine, but these are important points that readers should be aware of:
"This is an important topic and is something we have thought about. In several years of bloom sampling across the region (2019, 2022, 2023), we have seen that the suite of toxins produced by Pacific Arctic *A. catenella* is consistently dominated by gonyautoxin-4, neosaxitoxin, gonyautoxin-3 and saxitoxin (Lefebvre et al. 2022). Unfortunately, both gonyautoxin-4 and neosaxitoxin have low cross-reactivities with the ELISA test (<2%), but STX is picked up at 100% and gonyautoxin-3 at 23%. So, while the ELISA is likely underestimating the total amount of toxin in these samples, the overall consistency that we have observed in toxin profiles of regional *A. catenella* strains across years indicates that ELISA data are appropriate for assessing relative temporal trends in toxicity, such as the results reported in this study. We believe these STX quantifications are representative of prevalence and relative concentrations equally over the two decades of sampling."

This point should make it into the manuscript:
"The likely reason for the much higher prevalences of STX compared to DA is that *A. catenella* blooms have two pathways for initiation of toxic blooms; blooms advected into the Beaufort from the Chukchi and locally initiated blooms from the Barrow cyst bed."

The following points are important and should be understood by readers via main text or supplemental findings:
We originally ran Spearman rank correlations of DA and STX concentrations (ng/g) in bowhead feces with the sea surface temperature (SST) anomalies for the summer months (June, July, August, and September) preceding harvest. We found that summer SST anomalies prior to harvest were positively and significantly correlated with DA concentrations in whale feces, however, there were no significant correlations of summer SSTs with STX concentrations in whale feces. We then moved on to look at heat flux which correlated with both toxins. We agree with the statement that the heat flux seems to be more important in determining STX concentrations in bowhead feces compared to only SST anomalies. If requested, we can include the SST analyses and figures in the supplemental materials.

For example: DA toxicity in feces was the most strongly correlated with July summer SST anomalies ($\rho=0.35$, $p < 0.01$, Figure below). DA was also positively and significantly correlated with June ($\rho = 0.17$, $p=0.02$), August ($\rho=0.28$, $p < 0.01$), and September ($\rho = 0.25$, $p < 0.01$) SST anomalies. However, STX concentrations in bowhead feces were not significantly correlated with SST anomalies ($p>0.05$ for all months).

Fig: Spearman rank correlation results ($\rho=0.35$, $p < 0.01$) comparing domoic acid (DA) concentrations in bowhead feces with July sea surface temperature (SST) anomalies. Points represent the mean \pm standard error DA concentration (ng/g) and color represents how many whales were sampled at that temperature.

The following points are important and should be understood by readers via main text or supplemental findings:
We suggest that the weaker relationship for STX and OW, as well as the more muddled heat flux relationship of STX compared to DA is due to the higher prevalence of STX in Arctic food webs compared to DA. The lowest prevalence of STX in any year was 44% in whales compared to DA where there were years with 0% prevalence. The likely reason for the much higher prevalences of STX compared to DA is that *A. catenella* blooms have two pathways for initiation of toxic blooms; blooms advected into the Beaufort from the Chukchi and locally initiated blooms from the Barrow cyst bed. We have added the STX prevalence vs open water anomalies comparison to the supplemental section with fig. S4. Additionally, the local source of *A. catenella* bloom initiation from the Barrow cyst bed is also responsible for the differences in the relationship to heat flux compared to DA (with no cyst bed component).

The following point should be acknowledged in the manuscript:
Changes in the relationship between the HABs and zooplankton have affected by temperatures could have influenced the

abundance of toxins in whale feces.

It would be worth mentioning the potentially connectivity of Alexandrium blooms across these regions with the implication that some trends found here may be happening elsewhere

Referee #3

(Remarks to the Author)

I am satisfied with the revisions and recommend the manuscript be accepted for publication.

Nate Mantua
NOAA/NMFS Southwest Fisheries Science Center
Santa Cruz

We thank all three referees for thoughtful and helpful reviews!

Here we provide detailed responses to referee comments with reference to line numbers and tracked changes in the revised manuscript (pdf).

Referee #1 (Remarks to the Author):

In an emerging field of ecosystem sentinel science, this paper is a stand-out. I have recently co-authored two scoping reviews on the topic (Hazen et al. 2019, 2024), and this study rises above most of what has been published to date. The mechanistic links between physical environment, primary producers, and top predators are made so clearly, with large effect sizes, multiple lines of evidence, over decades of data collection. As a result this paper will be broadly interesting to those studying the effects of climate change on the oceans, marine mammals, harmful algal blooms, and physical/chemical/biological oceanography to name a few. The figures are well made, contain the appropriate information, and are visually appealing (though legibility would be improved on some, e.g., larger text for the axes in Fig. 5). I do not have the necessary expertise to evaluate the details of the oceanographic modeling and algal data extraction and analysis; so did not thoroughly review those methods; that said, nothing jumps out to me as problematic. All this said, I do have a few major comments below that would need to be well addressed before I can endorse the paper's publication in Nature.

Response: The authors would like to thank referee #1 for their thorough review of the manuscript and supportive comments. The referee provides excellent points that we will address below.

Major comments:

The work feels like parachute science to me. Of the 12 authors of the paper, only three are based in Alaska, and it seems like none are members of the native Alaskan community that were essential in the collection of the key samples (bowhead whale fecal samples) that would have been impossible to acquire otherwise. Why is that? I could be wrong and may be overlooking a Native American co-author, and please let me know if this is the case. I understand that co-authorships should not be awarded without merit, but I can only imagine the amount of work native harvesters had to do to make this paper possible. This deserves more than one sentence of credit in the Acknowledgments section as is currently the case. Please provide an explanation for this omission.

Response: We thank the referee for commenting on this important issue! There is nothing more important than being sensitive to the needs of the communities involved in research. This is definitely not a parachute science project. We have been collaborating with the North Slope Borough (NSB) and subsistence whaling communities for 15 years. This work started as a service to the NSB Department of Wildlife Management (NSB-DWM) as part of their bowhead health assessment program. The goals of the NSB-DWM are to preserve healthy wildlife populations, integrate traditional knowledge with scientific research, advocate for community interests, and secure diverse funding to support ongoing projects (like the bowhead health monitoring program). Our team at the Northwest Fisheries Science Center (NWFSC) developed a program called the Wildlife Algal-toxin Research and Response Network (WARRN-West) that

provides surveillance for the presence of algal toxins in marine mammals – both stranded and/or harvested for subsistence purposes along the west coast of North America (Beaufort Sea to Southern CA). Each year we received bowhead fecal samples from whales harvested for subsistence purposes from the NSB-DWM to analyze for the presence of algal toxins as part of health assessments for this marine resource that is essential to the nutritional, cultural, and economic well-being of coastal communities throughout NW Alaska. We analyzed samples free of charge and returned results each year to the community. This is an incredibly strong and mutually appreciated collaboration. As our collaboration persisted over years, I had the vision that this could be an incredible dataset to use to evaluate changes in algal toxin presence in food webs over time using the bowhead whale as an integrative food web sampler. We have shared all results directly with the NSB-DWM and the whaling community of Utqiagvik annually. This is a full collaboration that serves the Alaskan Native community and is appreciated for assessment of bowhead whale health. I originally asked if there was a whaling captain or other tribal sampler that would like to be included as an author. Authorship was not important as their primary concern is the continued health of the whales. We do have an author that leads the NSB-DWM bowhead whale health assessment program (Raphaela Stimmelmayer) and who works directly with the community to share results and obtain samples.

For a bit more on the bowhead whale harvest monitoring program: The program was established in Utqiagvik in 1972, by Inupiaq leadership and is the key program of the NSB-DWM. The NSB-DWM is the wildlife research arm for the North Slope Borough communities and through a blend of scientific research, Indigenous knowledge, and community leadership, the department's research and policy work supports and protects the North Slope's traditional ways of life with a special focus on the management of the bowhead whale and the bowhead whale hunt (see NSB DWM website; <https://www.north-slope.org/departments/wildlife-management/>). The harvest of bowhead whales for subsistence purposes by Alaskan Native hunters is regulated by annual quotas through the International Whaling Commission (IWC) since 1977. In cooperation with the Alaskan Eskimo Whaling Commission (AEWC) and permission from whaling captains from 11 whaling communities, bowhead whales are regularly measured and inspected (post mortem evaluation) by hunters, NSB-DWM staff (i.e., biologists, veterinarians) to assess the health status of landed whales and to collect a diverse array of tissue samples for baseline data on life history, natural diseases, and marine threats. Bowhead harvest sampling in Utqiagvik is an effort of the whole department and thus many, many people have been involved over this time period.

As I mentioned in my overview of the paper above, this work has the potential to be a paragon of what an ecosystem sentinel study can be. However, the ecosystem sentinel literature is not cited at all. I find this omission confusing. In addition to the two review papers I mentioned above (Hazen et al. 2019, 2024), others you may consider drawing in and citing that are particularly relevant are (Scholin et al. 2000, Moore 2008, Bossart 2011, Bargu et al. 2012, Backer and Miller 2016, Plön et al. 2024). The lack of any of this literature being cited and discussed in the paper is glaring. Develop the ecosystem sentinel framework more in the main text as it's the integrative concept that ties this interdisciplinary study together. Topics you may consider discussing, are bowheads an example of a leading or elucidating sentinel (sensu Hazen et al. 2019)?; how would you recommend the knowledge gained in this study be used monitor

for HABs in this region in the future? What does this work say about the future of ecosystem sentinel science? And so on.

Response: We thank the referee for suggesting that we include more ecosystem sentinel literature. We did have Scholin et al. 2000 already cited in the original submission, but it was definitely an oversight not to include a review reference to using top predators as sentinels as that is exactly what we have done with these bowhead whales. We have added the excellent review by Hazen et al. 2019 with reference to the clear definition of ecosystem sentinels, exactly as we have utilized bowhead whales in this study (integrate information throughout food web; sentinels of ecosystem's response to climate variability etc). The following sentence was added in the main text Lines 78-80 in the revised manuscript with Hazen et al. 2019 reference. *"These wide ranging baleen whales filter-feed throughout the water column, primarily on copepods (Calanus sp.) and krill (euphausiids)¹⁸, making them excellent sentinels for trophic transfer of algal toxins as it relates to climate variability over time ¹⁹."*

There is excellent discussion and contextualization of the relevant oceanography and the algal species' ecology in the paper. However, this is mirrored by a near-complete lack of the ecology of bowhead whales and calanoid copepods (and other small zooplankton), the whales' main prey. Why is this? It is never even stated that bowhead whales are there to feeding on zooplankton, and presumably ingest phytoplankton incidentally (or contained within their copepod prey?). In a general journal like Nature, you need to be more explicit about the basic ecology of your study species and cannot assume the reader will know this. All this context, with supporting references, is sorely needed.

Response: We appreciate the comment that this information is not clear in the manuscript. We hope that by adding the sentence for the previous request helps make it clearer that the whales are feeding on krill and copepods. The sentence (line 57-58) that says toxins are accumulated in zooplankton (primarily euphausiids, and copepods) that consume algae, should cover the toxin accumulation issue. We have added an additional reference (18) for bowhead prey (line 79). Sheffield, G. and J. C. George (2021). Chapter 28 - Diet and prey. The Bowhead Whale. J. C. George and J. G. M. Thewissen, Academic Press: 429-455.

I am curious how the low, medium, and high thresholds for DA and STX were chosen – was it from empirical cutoffs in the raw data, known risk thresholds to wildlife, something else? Why not just keep these quantitative concentration values as a continuous variable? More explanation is needed to justify this approach.

Response: We looked at the raw data and defined the bin sizes to ensure that each group has an adequate number of data samples, and the results are not sensitive to the bin sizes. We now state this in the revision, and the sample number of each group is now added to the caption of Fig. 2 (Lines 219 – 225).

Minor comments:

Line 48-49: remove vague language like “since time immemorial” with real information (e.g., “since at least 12000 BCE” or whatever is accurate) here and elsewhere in the paper.

Response: We changed “since time immemorial” to “for 5,000 years” (Line 49).

Fig 4B: Add error band, R2 value, or some way for the reader to easily evaluate the fit of this relationship.

Response: We originally had the correlation coefficient and p-value on the figure ($R = 0.77$, $p < 0.01$) in very small text. We have increased the font size of the R value and increased the size of the line and points on the figure. We have also added the 95% confidence interval on Figure 4b. The new figure 4b has been inserted in the edited manuscript as well as edits in the Fig. 4b legend shown in tracked changes (Lines 310-316).

Fig. 5: Make axes labels larger, hard to see now as they are too small.

Response: We have updated Fig. 5 with larger labels. The new figure 5 has been inserted into the edited manuscript.

Literature cited

Backer, L. C., and M. Miller. 2016. Sentinel Animals in a One Health Approach to Harmful Cyanobacterial and Algal Blooms. *Veterinary Sciences* 3:8.

Bargu, S., M. W. Silver, M. D. Ohman, C. R. Benitez-Nelson, and D. L. Garrison. 2012. Mystery behind Hitchcock’s birds. *Nature Geoscience* 5:2–3.

Bossart, G. D. 2011. Marine mammals as sentinel species for oceans and human health. *Veterinary Pathology* 48:676–690.

Hazen, E. L., B. Abrahms, S. Brodie, G. Carroll, M. G. Jacox, M. S. Savoca, K. L. Scales, W. J. Sydeman, and S. J. Bograd. 2019. Marine top predators as climate and ecosystem sentinels. *Frontiers in Ecology and the Environment* 17:565–574.

We added Hazen et al. 2019 (comprehensive review of ecosystem sentinels).

Hazen, E. L., M. S. Savoca, T. J. Clark-Wolf, M. Czapanskiy, P. M. Rabinowitz, and B. Abrahms. 2024. Ecosystem Sentinels as Early-Warning Indicators in the Anthropocene. *Annual Review of Environment and Resources* 49:573–598.

Moore, S. E. 2008. Marine mammals as ecosystem sentinels. *Journal of Mammalogy* 89:534–540.

Plön, S., K. Andra, L. Auditore, C. Gegout, P. J. Hale, O. Hampe, M. Ramilo-Henry, P. Burkhardt-Holm, A. M. Jaigirdar, L. Klein, M. K. Maewashe, J. Müssig, N. Ramsarup, N. Roussouw, R. Sabin, T. C. Shongwe, and P. Tuddenham. 2024. Marine mammals as indicators of Anthropocene Ocean Health. *npj Biodiversity* 3:1–9.

Scholin, C. A., F. Gulland, G. J. Doucette, S. Benson, M. Busman, F. P. Chavez, J. Cordaro, R. DeLong, A. De Vogelaere, J. Harvey, M. Haulena, K. Lefebvre, T. Lipscomb, S. Loscutoff, L. J. Lowenstine, R. Marin, P. E. Miller, W. A. McLellan, P. D. R. Moeller, C. L. Powell, T. Rowles, P. Silvagni, M. Silver, T. Spraker, V. Trainer, and F. M. Van Dolah. 2000. Mortality of sea lions along the central California coast linked to a toxic diatom bloom. *Nature* 403:80–84.

This reference Scholin et al. 2000 was included in the original submission.

Referee #2 (Remarks to the Author):

This study links the rapid onset of climate change in the Arctic Ocean to the expansion of harmful algal blooms (HABs) and subsequent transfer of algal toxins to planktivorous bow whales in this system near Alaska. The study is very highly interdisciplinary and brings together biologists, chemists, physical oceanographers, atmospheric scientists, and indigenous peoples who are engaged in long-term monitoring. Beyond its impressive breadth, the study also has a remarkably unique data set and reveals intriguing linkages between changing climate, HABs, and trophic transfer of toxins. And while there is some good evidence to support these linkages, there are also outstanding questions, of course.

Response: We thank referee #2 for their detailed and thorough review of the manuscript. We appreciate the positive and supportive comments. The referee makes very valuable comments that we address in detail below.

The Arctic Ocean is a large ecosystem. The mooring used to track temperatures, velocities, and heat flux is located at a fixed point 150 km east of Pt. Barrow. There was no indication of where the 205 whales were collected from for this study relative to the mooring site. Placing points on a map would be very helpful. The authors argue, these animals are integrating the water column, so detailing this information will help make this case, to some extent, by showing how far afield the trends persisted.

Response: This is a good point. We should have clarified that all bowhead whales were landed at Utqiagvik (formerly Barrow) and harvested within a 30-mile radius. We have added this information to the Methods under section “Bowhead whale fecal sample collection” (Lines 510-513) with the following reference (36) also added.

ASHJIAN, CARIN J., et al. “Climate Variability, Oceanography, Bowhead Whale Distribution, and Iñupiat Subsistence Whaling near Barrow, Alaska.” *Arctic*, vol. 63, no. 2, 2010, pp. 179–94. *JSTOR*, <http://www.jstor.org/stable/27821962>. Accessed 26 Mar. 2025.

Arguably, the most important figures in the paper is Figure 2b and 2c. Are there any statistics that can be applied to Figure 2b and 2c to identify the precise period when the groups are significantly different from each other instead of simply calling out the peak date?

Response: The standard errors are included in Figs 2b and 2c (shown as shading). This quantitatively indicates where the different groups are significantly different from each other (i.e., where the shadings do not overlap).

Figure 2b and 2c. Why is the heat flux so much larger for SXT compared to DA?

Response: They are actually not very different. To avoid such confusion, we changed the scales on the y-axis for both plots (3×10^{12} and 1.5×10^{12} ; Fig. 2b and 2c, respectively).

Figure 2b and 2c. Was there a reason for the specific bin sizes used?

Response: We defined the bin sizes to ensure that each group has an adequate number of data samples, and the results are not sensitive to the bin sizes. We now state this in the revision (179-180), and the sample number of each group is now added to the caption of Fig. 2 (Lines 222-224).

Whale feeding: What size particles do the whales feed on, precisely? If they consume copepods, might they directly consume concatenated HAB cells that could be on a similar size. Also, as what depths do they feed? How does this compare to where HABs occur?

Response: Bowhead whales primarily feed on copepods and euphausiids. We added a sentence and reference on bowhead diet (line 78-80). It is likely that some algal cells are also consumed incidentally during feeding on zooplankton. Bowhead whales are known to feed near Utqiagvik at depths ranging from shallow continental shelf (45m) to the deeper waters (>300m) of Barrow Canyon (Citta et al. 2015)(Sheffield and George 2021). Depths at which *Alexandrium* cells mainly occur are in the upper 25m (Anderson et al. 2021) and *Pseudo nitzschia* particulate DA measurements have been documented from the surface (~2 m) to chl-a max depths (20 - 40m) (Hubbard et al. 2023).

Citta, J.J., Quakenbush, L.T., Okkonen, S.R., Druckenmiller, M.L., Maslowski, W., Clement-Kinney, J., et al., 2015. Ecological characteristics of core areas used by western Arctic bowhead whales, 20062012. *Prog. Oceanogr.* 136, 201222. <https://doi.org/10.1016/j.pocean.2014.08.012>.

Sheffield, G. and J. C. George. 2021. Diet and Prey. Pages 429-456. In *The Bowhead Whale: Biology and Conservation* J. G. M. 'Hans' Thewissen and John Craig George (eds.), Academic Press. 668 pp. <https://doi.org/10.1016/B978-0-12-818969-6.00028-5>

Could toxins have degraded in the stored samples over this 20-year study? Is there a control for this? Given the temperatures have been rising over time (higher toxins prevalence in recent samples) and that older samples would be more likely to have degraded toxins (higher toxins prevalence in recent samples), does degradation of toxins need to be considered? Can it be discounted?

Response: This is an excellent question. We did have a control study. To address this issue, we performed toxin degradation studies for both DA and STX in Bowhead feces stored under various conditions. The data confirmed that degradation of toxins in raw feces with frozen storage conditions did not occur for either toxin up to 4 years in storage (total time tested). All bowhead samples in this study from 2010-2024 were analyzed within the year of sampling. Samples from 2004-2009 were analyzed within 5 years of frozen storage. We added this text in the Methods section (Lines 552-557).

We published these findings and added the following references to the manuscript;

Bowers, E.K., Stimmelmayer, R., Hendrix, A., Lefebvre, K.A., 2022. Stability of Saxitoxin in 50% Methanol Fecal Extracts and Raw Feces from Bowhead Whales (*Balaena mysticetus*). *Marine Drugs* 20(9), 547.

Bowers, E. K., R. Stimmelmayer and K. A. Lefebvre. 2021. Stability of Domoic Acid in 50% Methanol Extracts and Raw Fecal Material from Bowhead Whales (*Balaena mysticetus*). *Mar Drugs* 19(8).

The ELISA kits were used to measure toxins. The SXT ELISA has the advantage of reacting with all congeners, but its efficacy for doing so varies with each congener. How does its specificity for specific congeners match up with the congeners most prevalent in the strains of *A. catenella* prevalent in the Arctic? Does this need to be considered?

Response: Thank you for this question. This is an important topic and is something we have thought about. In several years of bloom sampling across the region (2019, 2022, 2023), we have seen that the suite of toxins produced by Pacific Arctic *A. catenella* is consistently dominated by gonyautoxin-4, neosaxitoxin, gonyautoxin-3 and saxitoxin (Lefebvre et al. 2022). Unfortunately, both gonyautoxin-4 and neosaxitoxin have low cross-reactivities with the ELISA test (<2%), but STX is picked up at 100% and gonyautoxin-3 at 23%. So, while the ELISA is likely underestimating the total amount of toxin in these samples, the overall consistency that we have observed in toxin profiles of regional *A. catenella* strains across years indicates that ELISA data are appropriate for assessing relative temporal trends in toxicity, such as the results reported in this study. We believe these STX quantifications are representative of prevalence and relative concentrations equally over the two decades of sampling.

Which direction was the heat flux? I see the value '125°T' but is that vertical, horizontal? N, S, E, W? I think this should be stated outright and should be discussed with regard to the relevance for the development of each HAB.

Response: We have clarified that the heat flux is horizontal. Also, 125°T (where °T = degrees true) represents the direction 125 degrees clockwise from true north; i.e., pointing southeastward. We have clarified this in the methods section under Mooring data and heat flux calculation section (Line 644).

Open water area were linked to DA prevalence. There is no information of how this related to SXT.

Response: This is a great point. We did run a similar analysis with STX prevalence groupings (years with 100% whales positive for STX and those with <100% whales positive for STX) and found similar results to DA prevalence, where open water anomalies in the Beaufort were significantly higher in summer months prior to harvest dates when 100% STX prevalence was documented. However, the magnitudes in differences between prevalence and open water anomalies for STX groups were much lower (new Fig. S4) compared to the OW anomalies documented between the DA groupings (Fig 4a). This may be due to how prevalent STX already is in the food web with the lowest prevalence of STX in any year being 44% in whales compared to DA where there were years with 0% prevalence. The likely reason for the much higher prevalences of STX compared to DA is that *A. catenella* blooms have two pathways for initiation of toxic blooms; blooms advected into the Beaufort from the Chukchi and locally initiated blooms from the Barrow cyst bed. We have added the STX prevalence vs open water anomalies comparison to the supplemental section with a figure (S4). We reference the supplementary figure on lines 293-294 in revised manuscript.

It is important to state whether toxin levels were correlated with just surface water temperatures; it appears they were not which is probably because the heat fluxes are more important but this is a case to further support this study.

Response: This is a good point. We originally ran Spearman rank correlations of DA and STX concentrations (ng/g) in bowhead feces with the sea surface temperature (SST) anomalies for the summer months (June, July, August, and September) preceding harvest. We found that summer SST anomalies prior to harvest were positively and significantly correlated with DA concentrations in whale feces, however, there were no significant correlations of summer SSTs with STX concentrations in whale feces. We then moved on to look at heat flux which correlated with both toxins. We agree with the statement that the heat flux seems to be more important in determining STX concentrations in bowhead feces compared to only SST anomalies. If requested, we can include the SST analyses and figures in the supplemental materials.

For example: DA toxicity in feces was the most strongly correlated with July summer SST anomalies ($\rho=0.35$, $p < 0.01$, Figure below). DA was also positively and significantly correlated with June ($\rho = 0.17$, $p=0.02$), August ($\rho=0.28$, $p < 0.01$), and September ($\rho = 0.25$, $p < 0.01$) SST anomalies. However, STX concentrations in bowhead feces were not significantly correlated with SST anomalies ($p>0.05$ for all months).

Fig: Spearman rank correlation results ($\rho=0.35$, $p < 0.01$) comparing domoic acid (DA) concentrations in bowhead feces with July sea surface temperature (SST) anomalies. Points represent the mean +/- standard error DA concentration (ng/g) and color represents how many whales were sampled at that temperature.

We did not include these analyses in the manuscript because the heat flux was a more sensitive metric and provides more mechanistic information.

Differences between SXT and DA. There was no investigation of OW area and SXT and one would presume this is because it did not exist. Similarly, while the differences in prior heat flux

among the binned toxin levels were clear and large for DA, they were smaller and more muddled for SXT. Together, this suggests that temperature changes and OW changes are more important for DA; why might that be?

Response: These are excellent points regarding differences in heat flux and OW relationships for DA and STX. Some of this was addressed earlier with the following response regarding STX and OW; *“We did run a similar analysis with STX prevalence groupings (years with 100% whales positive for STX and those with <100% whales positive for STX) and found similar results to DA prevalence, where open water anomalies in the Beaufort were significantly higher in summer months prior to harvest dates when 100% STX prevalence was documented. However, the magnitudes in differences between open water anomalies vs prevalence for STX groups were much less (Fig S4) compared to the OW anomalies vs prevalence documented between the DA groupings (Fig 4a).”* We suggest that the weaker relationship for STX and OW, as well as the more muddled heat flux relationship of STX compared to DA is due to the higher prevalence of STX in Arctic food webs compared to DA. The lowest prevalence of STX in any year was 44% in whales compared to DA where there were years with 0% prevalence. The likely reason for the much higher prevalences of STX compared to DA is that *A. catenella* blooms have two pathways for initiation of toxic blooms; blooms advected into the Beaufort from the Chukchi and locally initiated blooms from the Barrow cyst bed. We have added the STX prevalence vs open water anomalies comparison to the supplemental section with fig. S4. Additionally, the local source of *A. catenella* bloom initiation from the Barrow cyst bed is also responsible for the differences in the relationship to heat flux compared to DA (with no cyst bed component).

This study is very focused on the HABs and toxins in feces, but an important step between these two outcomes is the consumption of the HABs by zooplankton which are actually consumed by the whales. This merits consideration and discussion. How might changes in the relationship between the HABs and zooplankton have affected by temperatures and, in turn, affected the outcomes of this study?

Response: This is an excellent question. There are certainly impacts of temperature on species distribution. We are not able to identify species distribution changes for zooplankton in this study. However, we do know that warmer temperatures are directly related to HAB prevalence and toxin concentration in food webs (i.e. prey) for both zooplankton and whales. This study shows that those increases in food web toxin prevalence and concentration tracked using bowhead feces are quantifiably correlated with warmer ocean conditions.

Much of the prior work describing the dynamics of Alexandrium blooms, cells, and cysts and seeding of the (by D. M. Anderson and others) seem to have been focused on regions south of the Bering Strait and accounting for how blooms in that region seed the Chukchi Sea and, to lesser extent how the Chukchi Sea might seed the southern Beaufort Sea. Given this, it would seem a broadening of the implications of this study would have value. Might the same trends seen here hold for the Bering Sea? The (southern) Chukchi Sea? It would be nice to see the explanation of Figure 3 broadened to include outcomes for the whole region.

Response: This is an interesting point but beyond the scope of this study. Extending these analyses to the southern Chukchi or Bering Sea would require the consideration of completely different atmospheric patterns (e.g. the Aleutian Low and its complicated storm track), as well as a long-term biological dataset of similar quality to the bowhead fecal time series presented here. While we do not have the pieces for this at this time, our hope is that this study can motivate similar work in the future for these other regions.

Line 35: 'Negative' instead of 'Devastating'; less drama

Response: Done. Changed Devastating to "Negative" (Line 36).

Line 47: 'supplies' instead of 'security';

Response: At the editor's discretion, we would like to keep the term food security as it has a specific functional meaning for Tribal communities (Line 48).

Line 48: For 'millennia' instead of 'since time immemorial'; accurate quantification.

Response: Done. Changed Immemorial to "for 5,000 years" (Line 49).

Lines 64-67: Consider combining sentences and clarifying what was monitored and precisely why.

Response: Thank you for this comment to clarify the monitoring program. We have changed the text and include what is being monitored and why. See Lines 72-75 in revised manuscript.

Line 69: Is their diet exclusively zooplankton? What is known regarding their minimum particle size retention? Might they directly consume concatenated HAB cells?

Response: This is similar to a request above. Bowhead whales primarily feed on copepods and euphausiids. We added a sentence and reference on bowhead diet (line 78-80). It is likely that algal cells are also consumed incidentally during feeding on zooplankton.

Line 82: Increased risks or increase occurrence?

Response: Both. The increased "prevalence" term covers both increased occurrence and therefore, increased risks to wildlife.

Lines 85-88: This statement should be supported by references to publications

Response: Yes, good point. We have added a reference for cyst distribution in the Alaskan Arctic (reference 22; Line 97).

Lines 110-112: Seems like there should be references to cite for new genotype /populations of *Pseudo-nitzschia* that was 'previously recorded'; cite this previous record.

Response: We added this reference that describes Pacific genotypes of *P. seriata*: (28) Hubbard, K. A., Rocap, G. & Armbrust, E. V. 2008. Inter- and intraspecific community structure within the diatom genus *Pseudo-nitzschia* (Bacillariophyceae). *Journal of Phycology* 44:637–649. (Line 133)

Lines 110-113: Do *P. seriata*, *P. pungens*, and *P. obtusa* all make DA? All at high levels? Details with references here are needed.

Response: *P. seriata* is a particularly toxic species, while *P. pungens* and *P. obtusa* have also been shown to produce DA. We have added a reference (26) to lines 125 and 131, as well as text in tracked changes (Lines 133-137).

Lines 114-119: Are these hypotheses necessary? Seems clear

Response: We agree it seems clear to those who know about algal blooms. We thought they might be valuable for those who are not familiar with algal blooms. At the editor's discretion, we would like to leave these for lay readers if deemed appropriate.

Figure 1a: What does the horizontal distribution of the data represent in the plot?

Response: Good question. The horizontal distribution in Fig. 1a does not represent anything. This is an added jitter of the points so readers can see how many years had the different (or similar) toxin prevalence values. For example, there are 7 years with 100% DA prevalence, if there was no jitter added to the plot, the points would lie on top of each other. Even with some overlapping of points, the added violin plots visualize the distributions of toxin prevalence in our dataset. We have added the following to the Fig 1a caption (Lines 152-154): "*Horizontal distribution in points is an added jitter effect for effective visualization of similar toxin prevalences among years (i.e., points)*".

Figure 1a is a little small and unclear; red and green dashed lines are faint, hard to see.

Response: Yes, we agree that the figure is hard to see. We have updated Fig 1a and made the lines thicker and dashed instead of dotted for better visualization of data.

Fig 1C. Indicate in Fig 1b where Fig1C is located.

Response: We have added a box to panel B near the cyst bed to show the location of panel C, and adjusted the legend to reflect the location and orientation of the cross-section (lines 157-159).

Line 145: Velocity of what and in what direction? Horizontal? Vertical? More detail needed. Also, doesn't the direction of the velocity matter a lot with regard to stated hypotheses?

Response: Yes, thank you. We have now clarified that the mooring measures the horizontal velocity of the water, and that positive heat flux is directed southeastward along the main path of the current (Lines 172-177).

Line 149: Did the heat flux affect HAB toxin levels in bowhead whales or did it effect the HABs and food web/zooplankton?

Response: We are trying to convey that the HAB toxin levels found in bowhead whale feces are due to consumption of toxic zooplankton. This seems clear in the manuscript. The toxin levels in the zooplankton are a result of the HAB density and toxicity. The heat flux impacts the HAB growth (increases) and therefore adds more toxins to the food web. So, HAB toxin concentrations in bowhead whales are influenced by heat flux indirectly via trophic transfer from more toxic zooplankton.

Line 152: Why 20 days? Why not more or less. See discussion

Response: This is because the advective time from Barrow Canyon to the feeding area near the mooring site is approximately 20 days. This is stated in the manuscript (Lines 181, 211).

Line 156: “with the largest effect beyond 13 days prior”...The meaning of this statement is unclear.

Response: Thank you for pointing this out. We have now clarified that the high DA group is associated with the highest heat flux, with increasing difference from the other two groups beyond 5 days prior. (Line 185-186).

Line 161: “within 10 days prior, the high STX group (>50 ng/g) had the highest heat flux”. Misstated. The STX did not have a heat flux.

Response: Good point. We changed the text to say “within 10 days prior, the high STX group was associated with the highest heat flux,...” (Line 191).

Line 163-174: Might not the difference in heat flux time and HAB be longer for Alexandrium given it starts as a benthic cysts in sediments? Are these heat fluxes important for cysts?

Response: We suspect that the main reason for the difference is that the local *Alexandrium* cyst bed is closer to the mooring site, hence the travel time is shorter versus the advective time from the Chukchi Sea/Barrow Canyon. This is stated in the manuscript. (Lines 193-212).

Lines 204-204: “which were progressively weaker corresponding to times of fecal collections that contained higher toxin concentrations”. Awkward phrasing.

Response: Yes, thank you for pointing this out. It is awkward. We have changed the text (Lines 233-235).

Figure 3: Why are the regions considered for sea level pressure and wind velocity different? It would be useful to show the area of focus show in a and b in the plots for c and d.

Response: The broader area is considered for sea level pressure because of the large-scale atmospheric patterns (in this case, the Beaufort High). We followed the suggestion to indicate the enlarged region on panel (3c) with a blue rectangle.

The correlation between OW and temperature (Fig 4b) is not directly related to this study.

Open water area linked to DA prevalence: There needs to be a statement of how OW related to SXT

Response: Great comment. This comment is similar to one made earlier. To summarize; OW area anomalies are significantly greater during the summer months when there’s 100% STX prevalence in whales compared to years with <100% prevalence, however, the effect is smaller compared to similar analyses done with DA prevalence (see Fig 4a and Fig. S4). We have added Fig. S4 to show this relationship (Lines 915-933 in revised manuscript).

Line 266-307: Yes, the region is warming, but is this directly related to the toxins in the feces? Is there any direct correlation between the toxin in feces and temperature? Figure 5 is interesting, but seems like a distraction from the data being presented in the paper. Is absolute

temperature important or heat flux? If it is the heat flux, rather than absolute temperature, this provides a more nuanced story that gets eroded by simple plots of rising temperatures.

Response: These are important points to clarify. Yes, there is a relationship for SST and DA concentrations in bowhead feces. This relationship is not significant for STX. This is most likely due to the prevalences of DA and STX currently in the system. STX is much more prevalent than DA, making it harder to see annual differences related to ocean conditions. The heat flux metric is a much more sensitive metric. The warmer ocean conditions associated with the warming climate influence heat flux at the Barrow buoy and open water area for the entire Arctic. The data shown in figure 5 show this warming trend and decrease in open water area since 1900 with some variability. This is an important part of this study.

Referee #3 (Remarks to the Author):

This manuscript reports on innovative, compelling, and important research linking Arctic climate and ocean conditions to the presence and intensity of toxic algae found in fecal samples obtained from bowhead whales that were taken during aboriginal subsistence harvests from 2004-2022. Key results reported here show that a combination of environmental conditions that include warm ocean temperatures, large ice-free open water area, weak NE winds along Alaska's NW coast, and increased heat flux to toxic algae concentrations in whale fecal samples. Trends in open water and ocean temperature provide a link to Arctic warming linked with global warming. This work is likely to generate broad interest from scientists, resource managers, aboriginal communities in the Arctic, and the general public. The article is very well written and the methods, results, and discussion points should be easy to understand for a broad audience save for one relatively minor issue that I flagged regarding the description of method used to create Fig. 3. Overall I liked this manuscript a lot and recommend that it be accepted for publication pending minor revisions. I have a few general comments for the authors to consider below, followed by specific comments for given line numbers, figures or tables. I congratulate the authors on this engaging, timely, interesting and important contribution.

Response: The authors thank referee #3 for their comprehensive review of this manuscript and their supportive remarks. We appreciate the important comments provided by the referee and address them in detail below.

General Comments:

1. Were fecal samples collected from years prior to 2004 analyzed for presence of neurotoxins? If not, can they be analyzed to provide a longer historical context? It seems like samples from periods with consistently less open water/more extensive sea ice (like in the 1970s-early 1990s, shown in your Fig 5d) would be very valuable for documenting a shift to increased HAB activity in your study period that is characterized by mostly low sea-ice extent/high open-water area and warm SSTs (especially since 2007). That said, your study period does contain substantial inter annual variation in those factors, but that variation is around a lower-ice baseline.

Response: Unfortunately, samples are not available before 2004. There are some sporadic single samples from various years, but very few and not enough for any kind of comparisons.

2. This work has some parallels with a study of *A. catenella* blooms in Puget Sound that was done about 15 years ago. That work also searched for environmental factors associated with bloom events sampled over two decades, and identified a combination of atmospheric/hydrologic/oceanographic factors were at play. The approach also considered the daily evolution of these factors. One issue that the Puget Sound work brought to mind while reviewing your manuscript is the daily evolution of surface winds, and how the associated “weak NE wind events” may have varied over the course of your composite events and over recent years and decades. Have you tried, or considered, developing a daily NE wind event index, or a daily Beaufort High pressure index?

Response: The relevant wind systems do not vary on daily timescales.

A daily atmospheric forcing index could be based on SLP (180-120W, 70-80N), or an index of NE wind speed in the 69-71N, 170W-145W region. You could then (a) document the daily evolution of wind forcing associated with composite events (similar to what you’ve done with heat flux **Note: We did not compute a daily heat flux, we integrated back in time.**), and (b) summarize the annual frequency, intensity, and duration of “weak NE wind events” associated with toxic algae events from the fecal samples. I raise this issue because understanding the history of favorable atmospheric forcing will complement what you’ve done with the annual time series of open water area and SST, and it adds a critical piece of your mechanism needed to go back in time or forward in time with forecasts and future projections. The Puget Sound work is summarized by Moore et al (2009) and Moore et al. (2011). I realize that adding a deeper dive into the atmospheric forcing at this point is probably more than what you’d like to do with this manuscript, and I don’t think it is needed to bolster what is currently reported. But it would be a nice addition to what you have already done, and could be part of another manuscript.

Moore, S.K., N.J. Mantua, B.M. Hickey, and V.L. Trainer. 2009. Recent trends in paralytic shellfish toxins in Puget Sound, relationships to climate, and capacity for prediction of toxic events. *Harmful Algae*, doi:10.1016/j.hal.2008.1003.

Moore, S.K., N.J. Mantua, E.P. Salathé Jr. 2011. Past trends and future scenarios for environmental conditions favoring the accumulation of paralytic shellfish toxins in Puget Sound shellfish. *Harmful Algae*, In Press, Corrected Proof, Available online 15 April 2011, ISSN 1568-9883, DOI: 10.1016/j.hal.2011.04.004. <http://dx.doi.org/10.1016/j.hal.2011.04.004>

Response: Thank you for this suggestion for future work. These were impressive studies by Moore et al. in Puget Sound. We will definitely think about this idea for a future manuscript.

3. For Fig 3, it is not clear how the composite wind vector maps were created. Lines 200-1 state that “the wind composites correspond to 15 days and 10 days before each fecal sample was collected ...”. I take this to mean the maps depict daily-mean surface wind vectors at the stated time points. If these are single-day daily mean fields please specify that in the Figure caption.

Related to my comment 2 above, what are the typical time-scales of weak NE wind events associated with high toxicity samples (a few days, a week)?

Response: Thank you for pointing out this confusion. The weak northeasterly wind conditions are not “events” (in contrast to the strong northeasterly wind conditions which are storm events). Hence the timescale could be days to weeks. We now clarify in the revision that the wind composites represent an average over the 15-day and 10-day period prior to when each fecal sample was collected for the three DA groups and STX groups, respectively (results are not sensitive to the precise averaging period). (Lines 230-233).

Line-specific comments:

64: Were fecal samples from years prior to 2004 analyzed for presence of neurotoxins? If not, why not? It seems like samples from periods with consistently less open water/more extensive sea ice (like in the 1970s-early 1990s, shown in your Fig 5d) would be very valuable for documenting a shift to increased HAB activity.

Response: Unfortunately, there are no samples available for earlier comparisons.

163-4: This analysis provides compelling evidence linking recent history of ocean heat flux to HAB toxin concentrations in bowhead whale feces, and that warmer ocean temperatures are linked to higher HAB toxin loads in the food web. Do these events also involve other aspects of HAB bloom dynamics that follow patterns of enrichment, stratification, nutrient uptake, nutrient stress etc seen in other systems?

Response: This is an area that we are interested in and which will certainly be a focus of future work, but such a comprehensive analysis will involve the incorporation of historical hydrographic and nutrient datasets, ultimately requiring a separate study. Work on *Alexandrium* (Anderson et al. 2021, 2022, Fachon et al. 2025) and *Pseudo-nitzschia* (Hubbard et al. 2023) from recent field surveys in the region have shown that these HABs display temperature, salinity, and nutrient associations as well as vertical and lateral spatial patterns, but our understanding of regional bloom dynamics and the interactions of various factors is still developing as we compile in situ data on algal abundance. Given that temperature is a strong driver for HABs as well as Arctic regional dynamics, it is an appropriate focus for this study.

186: indicate sub-sample sizes (number of events) used for the different sub-group composites; do Fig 2 panels (b) and (c) indicate a “relaxation” of strong advection events in the few days prior to sampling? I’m just trying to understand the time-evolution of the composite events

Response: We have added the sub-sample sizes in the figure 2 caption. The panels in (b) and (c) do not indicate “relaxation”. We simply integrate in time backwards regardless of winds. Later we show the effect of winds (Fig. 3).

200-1: for the 10 and 15 day composites, are these multi-day windows centered on 15 and 10 days prior, or single-day daily mean fields?

Response: Excellent clarifying question. The wind composites represent an average over the 15-day and 10-day period prior to when each fecal sample was collected for the three DA groups and STX groups, respectively (results are not sensitive to the precise averaging period).

235: again, are these composites of daily data for a single day or some window of days centered on 15 and 10 days prior to fecal sampling?

Response: See response above.

271, 820, 847, 874: are OW area anomalies in units of 100,000 sq km? Please indicate on the y-axis label or figure captions and in the top row of Tables S3 and S7 for EMM Anomaly

Response: In the Methods section we explain how we calculated the OW area anomalies (Lines 664-666): “Anomalies of OW in the Beaufort Sea were calculated by dividing summer monthly (June, July, August, and September) OW areas (km^2) by their respective monthly mean baseline (1982 – 2011) values (Table S6).” Thus, there are no units because they cancel out, resulting in a relative proportion of OW during fecal sampling years (2004 – 2022) compared to the baseline years (1982 – 2011) value. We have added a dashed line in all OW anomaly plots representing the baseline OW areas (i.e., value of 1). We added the following text to the figure captions 4a, S3, and S4. *“OW anomalies are calculated by dividing monthly OW areas (km^2) by the monthly baseline areas (km^2), resulting in a unitless anomaly equating baseline OW area values to 1 (dashed black line).”*

We also added the following text to tables S3 and S7: *OW anomalies are calculated by dividing monthly OW areas (km^2) by the monthly baseline areas (km^2), resulting in a unitless anomaly equating baseline OW area values to 1”.*

291-4: Are there also long-term trends in June-July-Aug atmospheric forcing patterns (the Beaufort High, or associated multi-day events with weak NE winds along the Beaufort coast), or have those been unchanged while trends in warming ocean temps and sea ice loss are driven by thermodynamic processes? You can probably come up with a daily index of atmospheric forcing based on your composite SLP, or an index of NE wind speed in the 69-71N, 170W-145W region.

Response: In the present study, the wind component of our study has to do with the time period just prior to when the whales are feeding (so this could be more episodic). We did also find an interesting relationship between the Arctic Oscillation Index in May and DA prevalence in fall whales. When AO was negative in May and the SST anomaly in the Beaufort Sea in July was > 2 degrees, the prevalence of DA was 100% in fall harvested whales in all years with these conditions. We did not find long term trends in changing positive vs. negative AO over time. We did find that after a negative May AO, the June open water area anomalies are significantly higher than June OW anomalies after positive May AO years in the Beaufort Sea. These findings support our current conclusions but are not as mechanistically definitive as the relationships shown in the present study so we did not include them. It is a good suggestion for the future, to investigate potential predictive indices.

We thank these referees again for their excellent and helpful reviews of this manuscript!
Here is a point-by-point response to the final comments.

Referee #1:

We have added more information describing the NWFSC/NSB/Inupiaq collaboration in the Inclusion and Ethics statement section as suggested by referee #1. This was the only request from referee #1.

Referee #2:

To address the first comment of referee #2, we added the following text to the methods section under Bowhead whale fecal sample collection (now Lines 447-451).

“The whales are known to feed near Utqiagvik at depths ranging from shallow continental shelf (45m) to the deeper waters (>300m) of Barrow Canyon (Citta et al. 2015)(Sheffield and George 2021). Depths at which *Alexandrium* cells mainly occur are in the upper 25m (Anderson et al. 2021) and *Pseudo nitzschia* particulate DA measurements have been documented from the surface (~2 m) to chl-a max depths (20 - 40m) (Hubbard et al. 2023).”

For the second comment, we added the following text to the methods section under Quantification of algal toxins in fecal samples (now Lines 494 – 503).

“In several years of bloom sampling across the region during this project (2019, 2022, 2023), the suite of toxins produced by Pacific Arctic *A. catenella* has been consistently dominated by gonyautoxin-4, neosaxitoxin, gonyautoxin-3 and saxitoxin¹³. Unfortunately, both gonyautoxin-4 and neosaxitoxin have low cross-reactivities with the ELISA test (<2%), but STX is picked up at 100% and gonyautoxin-3 at 23%. So, while the ELISA is likely underestimating the total amount of toxin in these fecal samples, the overall consistency observed in toxin profiles of regional *A. catenella* strains across years indicates that ELISA data are appropriate for assessing relative temporal trends in toxicity, such as the results reported in this study. These STX quantifications are representative of prevalence and relative concentrations equally over the two decades of sampling.”

For the third comment, the following text is in the Implications for Arctic food webs final paragraph of the manuscript (Lines 205-214);

“Two separate sources for *A. catenella* blooms, 1) cells advected in surface waters from the Bering and Chukchi Seas, and 2) local germination of cysts from the cyst bed east of Pt. Barrow (Fig. 1b), explain the higher prevalence of STX observed in Beaufort Sea food webs compared to DA (Fig. 1a), while *Pseudo-nitzschia* species are more dependent on advective processes for introduction into the Beaufort Sea. Warmer ocean temperatures increase the rates of both *A. catenella* cell growth and cyst germination resulting in larger more toxic blooms (Fig. 1c; ⁴). Dangerously high STX concentrations have recently been documented in Arctic food webs¹³. In contrast, DA prevalence is lower than STX, and DA concentrations quantified in bowhead whale feces are considered low in terms of poisoning risks to bowhead whales.”

We also added more of an explanation in the heat flux section (now Lines 158 – 161).

For the fourth comment, we added a figure and information regarding the correlation between SST and toxin concentrations in the Extended Data section (Extended Data Fig. 4; and lines 158 - 161) as requested.

For the fifth comment, the point regarding the cyst bed contribution to heat flux results is described in the manuscript (now Lines 147 - 157) and we also added the following text in red below (now Lines 157 – 161).

“These results indicate a clear relationship between upper layer heat flux and HAB toxin concentrations in bowhead whales and confirm that warmer ocean conditions are linked to higher HAB toxin loads in the food web. Both advected vegetative *A. catenella* cells from southern waters and cells germinated from local benthic cyst beds, are potential sources for STX-producing blooms in the Beaufort Sea, while *Pseudo-nitzschia* lacks the benthic contribution as it does not produce cysts. This may explain why DA has a stronger relationship to the heat flux in the 20 days before bowhead harvest, which reflects the advective time from Barrow Canyon to the feeding area near the mooring site. By contrast, STX concentrations correlate with heat flux within 10 days of bowhead harvest. This shorter time frame is likely due to the contribution of the western Beaufort Sea cyst bed located closer to the feeding site east of Pt. Barrow, in addition to advected *A. catenella* and likely explains the higher prevalence and potential toxicity of STX in Arctic food webs compared to DA (Fig. 1b). Additional comparisons with standard Beaufort Sea summer SST anomalies and toxin concentrations revealed significant correlations for DA, but not for STX also likely due to the already higher prevalence of STX in the Beaufort Sea (Extended Data Figure 4).

For the sixth comment: “changes in the relationship between HABs and zooplankton are affected by temperatures could have influenced the abundance of toxins in whale feces” is valuable, but we did not study those changes. The validated point from this study is that temperature is directly correlated to toxin prevalence and concentration in whale feces and therefore prey.

For the final comment: We don’t want to overstate this because the oceanography is so important for this finding and the oceanography patterns are different at the other locations.

Referee #3: No additional comments to address.